# Proteomic analysis of young and old mouse hematopoietic stem cells and their progenitors reveals post-transcriptional regulation in stem cells

Balyn W Zaro[1,2†*], Joseph J Noh[1,2†], Victoria L Mascetti[1,2], Janos Demeter[3], Benson George[1,2], Monika Zukowska[1,2], Gunsagar S Gulati[1,2], Rahul Sinha[1,2], Ryan A Flynn[4], Allison Banuelos[1,2], Allison Zhang[1,2], Adam C Wilkinson[1], Peter Jackson[3], Irving L Weissman[1,2,5,6*]

[1]Institute for Stem Cell Biology and Regenerative Medicine, Stanford University School of Medicine, Stanford, United States; [2]Ludwig Center for Cancer Stem Cell Research and Medicine, Stanford University School of Medicine, Stanford, United States; [3]Baxter Laboratory, Department of Microbiology and Immunology and Department of Pathology, Stanford University School of Medicine, Stanford, United States; [4]Department of Chemistry, Stanford University, Stanford, United States; [5]Department of Developmental Biology and the Stanford UC-Berkeley Stem Cell Institute, Stanford, United States; [6]Department of Pathology, Stanford University School of Medicine, Stanford, United States

*For correspondence:
balyn.zaro@ucsf.edu (BWZ);
irv@stanford.edu (ILW)

†These authors contributed equally to this work

Competing interests: The authors declare that no competing interests exist.

**Abstract** The balance of hematopoietic stem cell (HSC) self-renewal and differentiation is critical for a healthy blood supply; imbalances underlie hematological diseases. The importance of HSCs and their progenitors have led to their extensive characterization at genomic and transcriptomic levels. However, the proteomics of hematopoiesis remains incompletely understood. Here we report a proteomics resource from mass spectrometry of mouse young adult and old adult mouse HSCs, multipotent progenitors and oligopotent progenitors; 12 cell types in total. We validated differential protein levels, including confirmation that Dnmt3a protein levels are undetected in young adult mouse HSCs until forced into cycle. Additionally, through integrating proteomics and RNA-sequencing datasets, we identified a subset of genes with apparent post-transcriptional repression in young adult mouse HSCs. In summary, we report proteomic coverage of young and old mouse HSCs and progenitors, with broader implications for understanding mechanisms for stem cell maintenance, niche interactions and fate determination.

## Introduction

Hematopoietic stem cells (HSCs) are responsible for persistent renewal of blood and immune cells throughout a lifetime. They have the ability to not only self-renew, but also differentiate into effector cells in response to physiological demands such as infection or bleeding (*Figure 1A*; *Spangrude et al., 1988*; *Baum et al., 1992*; *Seita and Weissman, 2010*). HSCs have broad-reaching therapeutic promise in regenerative medicine, immunological tolerance, genetic autoimmune diseases, hematologic malignances and inherited disorders of the blood system (*Weissman, 2015*; *Weissman, 2005*). HSCs are also the locus of disease-causative mutations in a number of relatively common blood diseases and leukemias. HSC clones sustaining several driver mutations inhibit differentiation, drive proliferation, block programmed cell death and phagocytosis, and outcompete

**Figure 1.** Workflow and validation of proteomics in various hematopoietic stem and progenitor cells. (A) Hierarchy of hematopoietic differentiation. Hematopoietic stem cells (HSCs) give rise to multipotent progenitors (MPPs). Fate commitment arises in the oligopotent progenitor (OPP) compartment: megakaryocyte/erythrocyte progenitors (MEPs), common myeloid progenitors (CMPs), common lymphoid progenitors (CLPs) and granulocyte/macrophage progenitors (GMPs). (B) Proteomic sample preparation workflow: (a) Bone marrow cells are isolated as single-cell suspensions, (b) stained with a panel of antibodies, (c) sorted by FACS, and (d) lysed. After normalizing protein amounts, (e) the lysate is digested and desalted, and (f) peptides are subjected to mass spectrometry analysis. (C) The number of proteins identified in each cell type (N = 6). Each segment represents new proteins discovered as a result of each additional replicate. (D) Principal component analysis of all replicates of all cell types. (E) Normalized cKit protein intensity. (F) Normalized Ly6d protein intensity. (G) Normalized Ki67 protein intensity. (H) Single sample gene set enrichment analysis (ssGSEA) for GO cell-cycle-associated genes. P-adj = 0.00002. Enrichment scores were averaged across replicates for each cell type. FDR = 0.05. All violin plots show only non-zero intensity values. N.D. = not detected in any replicate.

The online version of this article includes the following figure supplement(s) for figure 1:

**Figure supplement 1.** Representative sorting scheme for HSCs and MPPs.
**Figure supplement 2.** Panther Protein Class analysis of data compiled for each cell type.
**Figure supplement 3.** Protein intensity ratios of the housekeeping protein Hprt1 in stem and progenitor cells.
**Figure supplement 4.** One-dimensional PCA plots show which components are key drivers of segmentation between cell types and cell compartments.
**Figure supplement 5.** Relative detection of proteins used for FACS purification of cell types by flow cytometry (dark gray) and MS (light gray).
**Figure supplement 6.** Single sample gene set enrichment analysis for GO DNA Repair.

normal HSCs (*Tomasetti et al., 2017*; *Jan et al., 2012*; *Miyamoto et al., 2000*; *Jamieson et al., 2004*; *Rossi et al., 2008*; *Pang et al., 2013*; *Busque et al., 2018*). The initiating driver mutations in HSCs on their own can lead to clonal hematopoiesis of indeterminate potential (CHIP), which predisposes an individual to these blood diseases, as well as atherosclerosis, and affects a significant percentage of aging populations (*Jaiswal et al., 2017*; *Jaiswal et al., 2014*). Understanding the transcriptomics and the proteomics of normal HSCs and each step of differentiation should reveal how these variations lead to such a wide swath of human diseases. In addition, insofar as most or all tissues and organs maintain their numbers by tissue-specific stem cells, lessons learned by examining HSCs could inform similar processes in other tissues.

Functional transplant studies and/or phenotypic genetic knockout mouse models have been the cornerstones of our understanding as to how HSCs maintain stemness and determine their fate (*Weissman, 2015*). More recently, much of what is known about gene expression of HSCs and their progeny has been discovered through DNA microarray, bulk and single-cell RNA-sequencing and ATAC-sequencing experiments, but few proteomic investigations on purified mouse HSCs have been conducted (*Seita et al., 2012*; *Cabezas-Wallscheid et al., 2014*; *Galeev et al., 2016*; *Buenrostro et al., 2018*). However, mRNA detection is a readout of translational potential and not protein presence, and therefore understanding the proteomic profiles of these cell types would allow for deeper insight into stem and progenitor biology. Furthermore, it has been well documented that mRNA abundance and protein abundance are not always well correlated (*Gygi et al., 1999*; *Koussounadis et al., 2015*; *Liu et al., 2016*). mRNA translation studies in HSCs suggest multiple modes in the regulation of protein abundance, and therefore mRNA levels of genes of interest may be insufficient for determining protein levels in HSCs (*Buszczak et al., 2014*; *Signer et al., 2014*). Signer and co-workers have also recently reported increased sensitivity to protein misfolding and a restricted capacity for proteasomal turnover in the HSC compartment (*Hidalgo San Jose et al., 2020*). These data support a hypothesis whereby mRNA translation can be the vital regulatory step in HSC fate determination, and a direct measurement of the suite of proteins within each cell type during hematopoiesis can provide further insight into biological mechanisms at play.

Several groups have previously performed mass spectrometry or mass cytometry analysis on mouse and human HSCs and progenitor cells, but these studies have been either highly-focused, conducted on mixed populations or, in the case of mass cytometry, require antibodies towards proteins of interest (*Jassinskaja et al., 2017*; *Palii et al., 2019*; *Cabezas-Wallscheid et al., 2014*; *Amon et al., 2019*). Currently, there is an incomplete understanding of the proteome across the entirety of early hematopoiesis, including HSCs, MPPs (MPPa, b, and c) as well as OPPs (CMP, GMP, MEP, CLP). Closing this knowledge gap provides us with the potential to determine key players in biochemical processes critical to hematopoiesis and broadly in stem cell biology, to identify cell-specific surface proteins for improved purification and to discover therapeutic protein targets.

HSCs are very rare and often difficult to purify, presenting a formidable challenge for traditional biochemical methods of investigation (*Mayle et al., 2013*). Experiments requiring large amounts of highly-pure starting material, such as cell lysate, have been technically arduous. Recently the Mann laboratory reported the use of mass spectrometry instrumentation capable of increased sensitivity and improved proteomic coverage from no more than 200 ng (~7000 cells), dramatically improving the feasibility of performing large-scale proteomics studies on low-abundant cell types (*Meier et al., 2018*). With this instrumentation available, we sought to complement and elaborate upon current proteomic data with a comprehensive unbiased proteomics database characterizing the proteome throughout young adult and old adult mouse hematopoiesis. We report here the proteomes of highly-purified HSCs and their progenitors in young and old hematopoiesis as detectable with very low input and state-of-the-art, yet accessible, mass spectrometry technology. Our database has been validated through FACS and fluorescence microscopy experiments for proteins of interest. We identified a unique relationship between mRNA abundance and protein abundance exclusive to the HSC compartment. These data are organized into a resource that allows for researchers to understand how protein abundance is altered during young adult and old adult HSC differentiation. This approach can be applied beyond HSCs to other rare cell types, including many types of stem and progenitor cells, where mass spectrometry technology in tandem with RNA-sequencing has yet to be applied to highly-purified samples.

## Results

### Optimization of sample preparation for low numbers of rare cells

Critical to our ability to deeply characterize the proteome throughout early mouse hematopoiesis was the development of a method by which to efficiently purify and process samples for mass spectrometry analysis from samples prepared with approximately 50,000 cells. To this end, we created a workflow whereby cells were purified by FACS using three sorting panels allowing for the isolation of 12 cells types: young adult and old adult HSC (Lin⁻, cKit⁺, Sca1⁺, CD34⁻, CD150⁺, Flt3⁻), MPPa (Lin⁻, cKit⁺, Sca1⁺, CD34⁺, CD150⁺, Flt3⁻), young adult and old adult MPPb (Lin⁻, cKit⁺, Sca1⁺,

CD34$^+$, CD150$^-$, Flt3$^-$), young adult and old adult MPPc (Lin$^-$, cKit$^+$, Sca1$^+$, CD34$^+$, CD150$^-$, Flt3$^+$), young adult CLP (Lin, cKit$^{lo}$, Sca1$^{lo}$, Flt3$^+$, IL7Rα$^+$), young adult CMP (Lin$^-$, cKit$^+$, Sca1$^{lo/-}$, CD34$^{med/hi}$, CD16/32$^{-/lo}$), young adult MEP (Lin$^-$, cKit$^+$, Sca1$^{lo/-}$, CD34$^-$, CD16/32$^{-/lo}$, CD150$^+$) and young adult GMP (Lin$^-$, cKit$^+$, Sca1$^{lo/-}$, CD34$^{hi}$, CD16/32$^{hi}$) (*Figure 1A* and *Figure 1—figure supplement 1*). Cells were purity sorted into FACS buffer (2% fetal bovine serum in PBS) and washed twice with PBS to remove any remaining serum prior to storage. Samples for mass spectrometry analysis were prepared with a minimum of 50,000 cells (*Figure 1B*). Since multiple sorts were required to obtain the requisite cells per biological cohort, we pooled samples and maintained equal contributions from each mouse for each cell type. We also normalized the amount of lysis buffer used with respect to cell number. A commercially-available mass spectrometry sample preparation kit was utilized to minimize sample loss and strengthen reproducibility and proved critical to our efforts (*Figure 1B*). For all young and old adult mouse HSCs, at least three biological cohorts of mice were utilized for each cell type, and the sample was run in technical duplicate with 200 ng (~7000 cells) of loading material per replicate. In total, six replicates were acquired for each of the eight young adult stem and progenitor cell types and old mouse HSCs, and four replicates for each of the three old MPPs were utilized. We included both biological and technical replicates to account for limitations in detection using mass spectrometry analysis. Despite utilizing state-of-the-art equipment which significantly improves issues of resolution, sensitivity and speed, there still may be scenarios where low-abundance peptides or peptides non-amenable to ionization are not well detected. By performing multiple technical replicates on the same sample, we ensure detection of as many proteins as possible (*Liu et al., 2004*). Each individual replicate was processed through Byonic software as an individual dataset. After performing mass spectrometry analysis in sextuplicate, we saw diminishing returns on additional replicates in the context of new protein discovery, minimizing the likelihood that differences in proteomic diversity are artifactual or a result of differential data quality (*Figure 1C*).

## A database of proteins expressed by rare cell types

In order to generate a repository of proteins present in HSCs and their progenitors that are detectable by mass spectrometry, we divided each individual protein raw intensity value by the total intensity detected for each technical replicate and multiplied by 1 million for ease of analysis (*Supplementary file 1*, Table 1). An average of non-zero values was taken for each gene within each cell type for global analyses (*Supplementary file 1*, Table 2). Across all cell types we detected a total of 7917 genes encoding proteins expressed and detectable in HSCs and their progenitors (*Supplementary file 1*, Tables 1 and 2). The adult HSC compartment had the least protein diversity, with 4030 proteins detected (*Figure 1C*). These values reflect the total number of unique proteins detectable by mass spectrometry. We observed a general trend in increased protein diversity during the differentiation process, with MPPs and OPPs expressing larger numbers of distinct proteins compared to HSCs (*Figure 1C*). This result differs from transcriptomic reports where HSCs present increased mRNA diversity in the stem cell compartment compared to their progeny (*Ramos et al., 2006*). To validate the quality of our coverage across all cell types analyzed, we performed PANTHER gene list analyses for both protein class (*Figure 1—figure supplement 2*; *Mi et al., 2019a*; *Mi et al., 2019b*). The percentages of classes of proteins detected were consistent across all cell types, indicating that samples were reproducibly processed and that no large subsets of proteins, by both class and function, were noticeably absent. To validate that differential proteomic expression profiles were not predominantly due to global differences in protein detection across cell types, we analyzed the abundance of the housekeeping protein Hprt1 (*Figure 1—figure supplement 3*). Protein levels were consistent across all cell types characterized. Principal component analysis (PCA) revealed exquisite distribution of adult hematopoietic stem and progenitor cells (*Figure 1D*, *Supplementary file 1*, Table 3). Excitingly, each component played a distinct role in separating cell types from one another: component 1 isolates HSCs from all other cell types, whereas component 2 separates stem and multipotent progenitors from the more-committed oligopotent progenitor compartment (*Figure 1D*, *Figure 1—figure supplement 4* and *Supplementary file 1*, Table 3).

## Characterization of HSC and progenitor proteomes

With our database generated, we next validated detection of known markers of stem and progenitor cells (*Figure 1E* and *Figure 1—figure supplement 5*). As expected, cKit was detected across all cell

types with levels high in HSCs, MPPs and lower in CLPs and GMPs. CD150/Slamf1 abundance was exclusive to HSC and MPPa compartments, an attestation to the purity of these sorted samples (*Figure 1—figure supplement 5*). Ly6d, a marker of early B-cell progenitors, was uniquely detected in the CLP compartment (*Figure 1F*; *Ghaedi et al., 2016*; *Inlay et al., 2009*). The cell-cycle associated protein Ki67 was lowly detected in the HSC compartment with a steady increase in abundance in the MPP compartment and highest in the OPPs (*Figure 1G*). We exclusively detected Flt3 in MPPc and CLP, CD16/32 in GMP and (very lowly) CMP, and IL7Ra in CLP (*Figure 1—figure supplement 5*). We also compared the relative median fluorescent intensities of surface markers as detected by FACS to their relative abundance by mass spectrometry (*Figure 1—figure supplement 5*). Similar to previous studies by Trumpp and co-workers, we were not able to detect Sca1 in any of our datasets (*Cabezas-Wallscheid et al., 2014*). Levels of CD34 detected in the HSC compartment were the only surprising results in our analysis, but Trumpp and co-workers have also detected variable levels of CD34 in the HSC compartment across their replicates. However, in both datasets, coverage of the protein is extremely low, and all other quality-control parameters suggest robustness and purity of the samples. Since neither approach included isolation of proteins based off of cellular localization, the levels of CD34 detected may be attributed to intracellular CD34 or perhaps CD34 antibodies are incapable of detection of CD34 in HSCs due to differential post-translational modifications and/or structural variation resulting in differential antibody binding.

To characterize enriched pathways, we performed geneset enrichment analyses (FDR = 0.05). Proteins associated with cell cycle and DNA damage repair were significantly less abundant in HSCs compared to progenitor cells, and both processes have been shown to be dramatically reduced in the HSC compartment (*Figure 1H* and *Figure 1—figure supplement 6*; *Nijnik et al., 2007*; *Pietras et al., 2011*; *Rossi et al., 2007a*; *Rossi et al., 2007b*). More specifically, it has previously been shown that quiescent HSCs accumulate DNA damage and old adult mouse HSCs have both nuclear phospho-γH2AX and DNA breaks by comet assay. However, bringing G0 HSCs into cell cycle in vitro leads to upregulation of many DNA repair pathways in G1 prior to entry into S phase and is sufficient to rescue most if not all HSCs (*Beerman et al., 2014*; *Rossi et al., 2007a*; *Rübe et al., 2011*). Given that approximately 80–90% of HSCs are quiescent, we anticipated lower levels of DNA repair-associated response in our dataset (*Rossi et al., 2007b*; *Sudo et al., 2000*; *Tesio et al., 2015*). For example, the double-strand DNA repair protein Rad51 is not detected in our HSC proteomics data but is detected in all other cell types. (*Supplementary file 1*, Tables 1 and 2; *Beerman et al., 2014*).

## Validation of differential abundance of proteins of interest

To validate differentially-detected proteins using non-mass spectrometry techniques, we performed FACS analysis and fluorescence microscopy. The endothelial surface adhesion molecule (Esam) has previously been shown to be highly expressed by HSCs (*Ishibashi et al., 2016*; *Ooi et al., 2009*; *Yokota et al., 2009*). In our dataset, Esam levels were very high in HSCs and MPPas with decreased abundance in MPPbs and no detection for the remaining cell types (*Figure 2A*). This result was recapitulated in flow cytometry analysis of Esam levels, further supporting the quality of the proteomics dataset. (*Figure 2A* and *Figure 2—figure supplement 1*). We also validated differential abundance of the regulatory glycolytic enzyme phosphofructokinase (Pfkl) by fluorescence microscopy (*Figure 2B and C*). Average Pfkl levels were decreased in the HSC compartment compared to MPPa and MPPb.

We noted that DNA-methyl transferase 3a (Dnmt3a) was not detected in any of our HSC replicates but was well detected in MPP and OPP populations (*Figure 2D* and *Supplementary file 1*, Tables 1 and 2). This finding was further validated by microscopy where freshly-sorted HSCs or a mixed population of stem and progenitor cells (LSK: Lin⁻, Sca1⁺, cKit⁺) were stained with anti-Ki67 and anti-Dnmt3a. HSCs were both Ki67 negative and Dnmt3a negative compared to LSK cells that were positive for both proteins (*Figure 2E*). Mutations in Dnmt3a expression have been implicated as disease-initiating mutations in hematologic malignancies and are among the most common mutations found in disease pathologies, including pre-AML mutations in HSCs (*Corces-Zimmerman et al., 2014*; *Jan et al., 2012*; *Ley et al., 2010*; *Yang et al., 2015*) and in CHIP (*Jaiswal et al., 2017*; *Jaiswal et al., 2014*). Dnmt3a's role in HSC biology has also been well studied (*Challen et al., 2012*; *Hu et al., 2015*; *Jeong et al., 2018*; *Tadokoro et al., 2007*). Self-renewal is perpetual in the absence of Dnmt3a and expression is required for differentiation (*Challen et al.,*

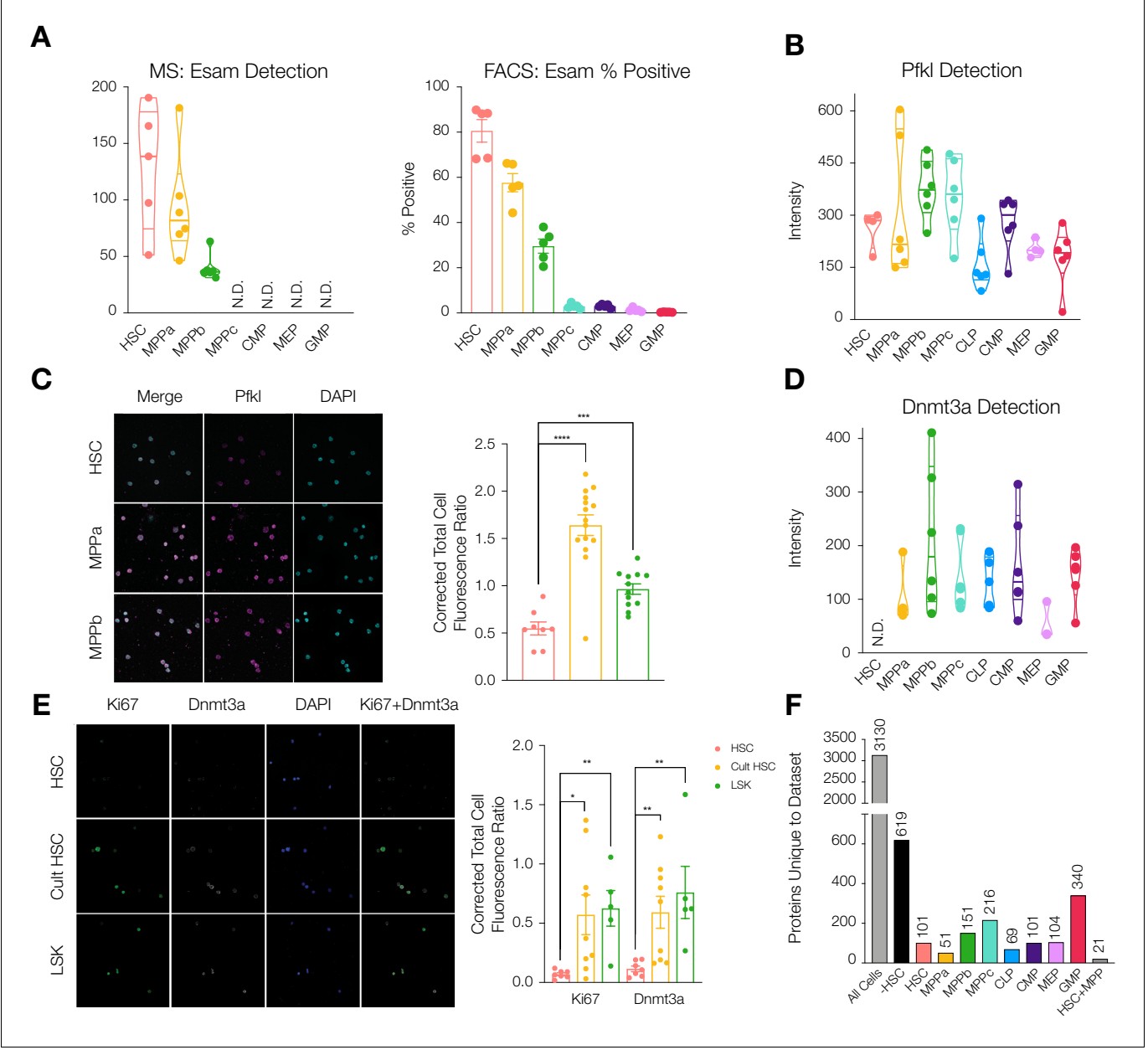

**Figure 2.** Differential protein levels throughout early hematopoiesis. (A) Normalized Esam protein intensity (left) and % Esam+ cells as determined by flow cytometry analysis (right). N = 5 mice (three male, two female). (B) Normalized Pfkl protein intensity values. (C) Fluorescence microscopy of HSC, MPPa and MPPb stained with anti-Pfkl (left) and Corrected Total Cell Fluorescence (CTCF) ratio of Pfkl/DAPI (right). N = 5 mice (three male, two female). P-values: ***=0.0002, ****<0.0001. (D) Normalized Dnmt3a protein intensity values. (E). Fluorescence microscopy of fresh HSCs (HSC), cultured HSCs (Cult HSC) and stem and progenitor cells Lin-, Sca1+, cKit+ (LSK) stained with anti-Ki67 and anti-Dnmt3a (left) and CTCF ratio of Ki67/DAPI and Dnmt3a/DAPI (right). N = 5 mice (three male, two female). P-values: Ki67: *=0.0202, **=0.0013 Dnmt3a: HSC vs Cult HSC **=0.0080, HSC vs LSK **=0.0057. F. Number of proteins uniquely detected in each subset of cell type(s). −HSC: proteins detected in all cell types except HSCs. HSC+MPP: proteins detected in HSCs and MPPs. All violin plots show only non-zero intensity values. N.D. = not detected in any replicate. Fluorescence was quantified using ImageJ.

The online version of this article includes the following figure supplement(s) for figure 2:

**Figure supplement 1.** FMOs and gating strategy for ESAM staining.

**Figure supplement 2.** Enrichment ratios between HSCs *vs.* MPP1 or MPPa (log2) for Igf2bp2 and Hmga2.

**Figure supplement 3.** Protein expression of Igf2bp2 and Hmga2 in young and old adult mouse HSCs and progenitors.

**Figure supplement 4.** Overlap of proteomic dataset compared to Cabezas-Wallsheid et al. and unique proteins detected in each study.

*2012*). DNA methylome analysis has shown that in the absence of Dnmt3a, genes promoting self-renewal are not repressed, therefore preventing differentiation (*Challen et al., 2012*). We reasoned that perhaps Dnmt3a protein abundance would increase in HSCs moving out of G0 and into cell cycle. To this end, HSCs were sorted and cultured under media conditions promoting cell cycle (*Wilkinson et al., 2019*). Fluorescence microscopy revealed that cultured HSCs have increased levels of Ki67 and Dnmt3a (*Figure 2E*). Our findings support a scenario where Dnmt3a protein is not present in G0 quiescent HSCs but is accumulated as HSCs enter cell cycle in order to silence stem-associated genes and enable HSC differentiation into multipotent or oligopotent cells. Given the sensitivity of HSCs to perturbations in protein synthesis and turnover, we further hypothesize that non-cycling HSCs likely do not synthesize Dnmt3a rather than synthesize the protein only to rapidly degrade it (*Hidalgo San Jose et al., 2020*; *Signer et al., 2014*).

In addition to our own independent validation, we compared our findings to previous literature reports. Insulin-like growth factor one receptor (Igf1r) has been shown to be undetectable by single-cell staining in the HSC compartment compared to MPPs (*Venkatraman et al., 2013*). Similarly, Igf1r is not detected in our young adult HSC mass spectrometry data but is detectable in the MPPs (*Supplementary file 1*, Tables 1 and 2). Lin, Goodell and co-workers have shown that the proliferation-associated protein CD81 is found on HSCs that have moved into cycle (*Lin et al., 2011*). In our studies, CD81 was not detected in young adult bone marrow-resident HSCs, which appear to have a more quiescent signature, but was found in all other cell types analyzed (*Supplementary file 1*, Tables 1 and 2).

## Characterization of proteins uniquely absent/detected by cell type(s)

Over 40% of proteins were detected across all cell types, 3130 proteins in total (*Figure 2F*). With deep proteome coverage and analysis of progenitor populations, we were also able to identify additional proteins like Dnmt3a uniquely absent or uniquely detected in a single-cell type (*Figure 2F* and *Supplementary file 1*, Table 4). For example, 619 proteins were absent in the HSC compartment but were found in all other cell types. In investigating uniquely-expressed proteins, particularly the 340 proteins exclusively found in the GMP compartment, we identified GMP-specific proteins including CCAAT-enhancer-binding protein ε (Cebpe), Adhesion G Protein-Coupled Receptor G3 (Adgrg3) and Membrane-spanning 4-domains, subfamily A, member 3 (Ms4a3), all of which have previously been reported to be associated with macrophage and granulocyte-specific lineage commitment (*Supplementary file 1*, Tables 1-3; *Goardon et al., 2011*; *Hsiao et al., 2018*; *Ishibashi et al., 2018*; *Nakajima et al., 2006*). In the HSC compartment, Igf2bp2 was detected as a uniquely-expressed protein, as has previously been identified as differentially expressed by HSCs compared to MPP1s (*Figure 2—figure supplements 2* and *3*; *Cabezas-Wallscheid et al., 2014*). 619 proteins were detected in all other cell types besides HSCs (*Figure 2F*). While we cannot assess the biological consequences of the absence of each of these proteins, the large number of uniquely absent proteins in the HSC compartment prompted further investigation as to potential mechanisms behind this attenuation of diversity.

## Comparison to previous mass spectrometry datasets

We compared our HSC (Lin⁻, cKit⁺, Sca1⁺, CD34⁻, CD150⁺, Flt3⁻) and MPPa (Lin⁻, cKit⁺, Sca1⁺, CD34⁺, CD150⁺, Flt3⁻) data to that of Trumpp and co-workers' comparative proteomics data which was focused exclusively on comparative studies between HSC (Lin⁻, Sca1⁺, cKit⁺, CD34⁻, Flt3⁻, CD48⁻, CD150⁺) and MPP1 (Lin⁻, Sca1⁺, cKit⁺, CD34⁺, Flt3⁻, CD48⁻, CD150⁺) (*Supplementary file 1*, Table 5) (*Cabezas-Wallscheid et al., 2014*). Across both datasets, a total of 6466 proteins were detected in HSCs and/or MPPas/MPP1s, with 70.43% overlap (*Figure 2—figure supplement 4*). Of the 49 differentially abundant proteins identified by Cabezas-Wallscheid et. al. that were also detected in our datasets, 37 were consistently differentially abundant across both experimental methods (at least 2-fold more frequently detected in HSC or early progenitor in our data, or denoted as differentially expressed per Cabezas-Wallscheid et. al.). Discrepancies in detection for other proteins could be due to a multitude of reasons, including different experimental methods (comparative vs. shotgun, tagged vs. label-free, sorting schemes, instrumentation, data analysis and statistical methodology). With most of these data consistent across both experiments, such as HSC-enriched abundance of Igf2bp2 and Hmga2, our additional cell type coverage allows for a deeper

resolution into differential and total detection levels throughout the hematopoietic tree (*Figure 2—figure supplements 2* and *3*; *Nishino et al., 2008*; *Nishino et al., 2013*). For example, Igf2bp2 is exclusively detected in HSCs in our datasets across all cell types, whereas Hmga2 is simply more abundant in the HSC compartment (*Figure 2—figure supplement 3*).

## Characterization of old adult HSC and MPP proteomes

Blood formation during aging is marked by a myeloid bias and higher frequency but lower engraftment per HSC transplanted (*Jaiswal et al., 2014*; *Morrison et al., 1996*; *Pang et al., 2011*). However, to the best of our knowledge, no proteomic experiments have characterized protein abundance changes in the HSC and MPP compartments during aging by mass spectrometry. Using our sort schemes and sample preparation methods, we purified and processed HSCs and MPPs from mice no less than 24 months of age (*Figure 1* and *Figure 1—figure supplement 1*). Data analysis revealed detection of 5434 proteins in old mouse HSCs, a 35% increase in protein diversity compared to young adult mouse HSCs, with comparable protein numbers detected across the young and old adult MPP compartments (*Figure 3A*, and *Supplementary file 1*, Tables 1 and 2). PCA analysis demonstrated the high similarity between young and old adult mouse HSCs as compared to progenitor cells but also important differences across both component 1, where old mouse HSCs

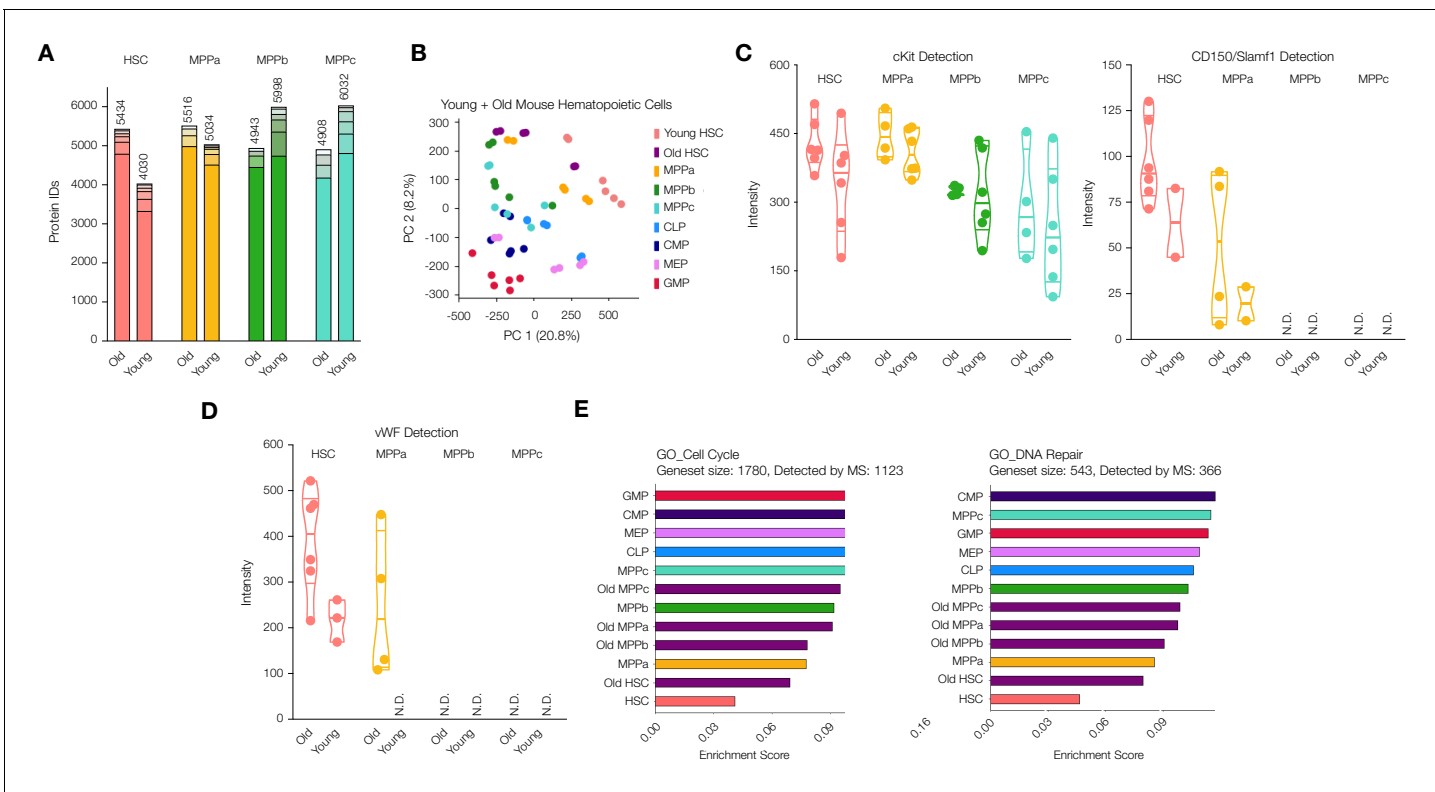

**Figure 3.** Proteomic comparison between young and old mouse HSCs and MPPs. (**A**) Total number of proteins identified across experimental replicates for old HSCs (N = 6) and old MPPs (N = 4) in comparison to young adult mouse HSCs and MPPs (N = 6). Each segment represents new proteins discovered as result of each additional replicate. (**B**) Principal component analysis of all replicates of all young adult cell types and old mouse HSCs. (**C**) Protein intensity values for known markers of stem and early progenitor cells, cKit, and CD150. (**D**) Protein intensity values of von Wilebrand factor (vWF). (**E**) ssGSEA of GO Cell Cycle and DNA Repair-associated genes including young and old adult mouse HSCs and progenitors. P-adj = 0.00002, 0.00015 and 0.00002, respectively. Enrichment scores were averaged across replicates for each cell type. FDR = 0.05 All violin plots show only non-zero intensity values. N.D. = not detected in any replicate.

The online version of this article includes the following figure supplement(s) for figure 3:

**Figure supplement 1.** One-dimensional PCA plots show, which components are key drivers of segmentation between cell types and cell compartments.

**Figure supplement 2.** Protein abundance of Ki67 in young and old adult mouse HSCs and progenitors.

**Figure supplement 3.** Protein abdundance of the age-associated protein Itgb3 in young and old adult mouse HSCs and progenitors.

lose the distinctness of young adult mouse HSCs, and component 2, where old mouse HSCs to occupy a unique protein signature in comparison to both young adult stem and progenitor compartments (*Figure 3B* and *Figure 3—figure supplement 1* and *Supplementary file 1*, Table 6). To this end, we also generated a list of proteins detected in the old adult mouse HSC compartment in at least three replicates that are either not detected in young adult mouse HSCs or are in the top 2.5% of fold-change between old vs. young intensity ratios (*Supplementary file 1*, Table 7). We believe this list to be a summary of high-confidence proteins that are more detectable in old HSCs compared to young.

As expected, cKit was consistently detected in all four old cell types while CD150/Slamf1 was found only in HSC and MPPa compartments exclusively (*Figure 3C*). cKit levels were on average higher in the old cells compared to their young adult counterparts, which has also been observed by others (*Figure 3C* and *Supplementary file 1*, Tables 1 and 2; *Beerman et al., 2010*; *Mann et al., 2018*). Earlier FACS separation of CD150$^{hi}$ and CD150$^{lo}$ HSCs revealed CD150$^{hi}$ HSCs are myeloid biased, and this sub-population increases most-dramatically in old mice (*Beerman et al., 2010*). Our proteomics data revealed similar variations in CD150 levels, with the range lowest in young adult mouse HSCs (*Figure 3C* and *Supplementary file 1*, Tables 1 and 2). Additionally, Ki67 detection was still lower in the old HSC compartment compared to that of young adult and old downstream progenitors, although more frequently detected than the young adult HSC compartment (*Figure 3—figure supplement 2*). We also detected an increase in Ki67 abundance in old MPPbs compared to young adult MPPbs.

Given the multitude of functional studies that have been conducted to identify genes implicated in HSC fate determination and stemness, we were interested to compare our findings of differentially expressed proteins during young vs. old adult hematopoiesis with previous reports. von-Willebrand Factor (vWF) is associated with myeloid and platelet biases during hematopoiesis, and we also detected increased vWF in old mouse HSCs and MPPas (*Figure 3D*; *Grover et al., 2016*; *Mann et al., 2018*; *Pinho et al., 2018*; *Sanjuan-Pla et al., 2013*). Integrin surface proteins are critical in mediating a pro-inflammatory response that can elicit bias in HSC fate determination, and such proteins are also well documented to increase during aging in addition to inflammatory events (*Gekas and Graf, 2013*; *Haas et al., 2015*; *Mann et al., 2018*; *Pang et al., 2011*). We detected Itga2b (CD41) in old mouse HSCs, which has been demonstrated to induce a myeloid bias, but, similar to Mann et. al., we did not see a significant difference between young and old adult (*Supplementary file 1*, Tables 1 and 2) (*Gekas and Graf, 2013*; *Mann et al., 2018*). However, levels of the complementary signaling molecule Itgb3 (CD61) increased in the old compartments of HSCs, MPPas and MPPbs, as described previously by Regev, Baltimore and co-workers (*Figure 3—figure supplement 3*; *Mann et al., 2018*). Additionally, we performed gene set enrichment analysis and identified that, like young adult mouse HSCs, old mouse HSCs had lower levels of abundance of proteins associated with cell cycle and DNA damage repair (*Figure 3E*). However, compared to young adult mouse HSCs, cell cycle and DNA damage repair-associated proteins were more enriched in the old cells.

## mRNA abundance comparison

Globally, mRNA expression and protein abundance are not well correlated across yeast and higher eukaryotes (*Liu et al., 2016*). We were interested to determine if there were changes in the relationship between mRNA and protein during hematopoiesis—both broadly across the proteome as well as for specific proteins of interest. Bulk mRNA sequencing was conducted from young adult mouse HSCs, MPPas, MPPbs and MPPcs as described previously (*Supplementary file 1*, Table 8; *Moraga et al., 2015*). The diversity of mRNAs was similar across cell types despite much lower protein diversity in HSCs (*Figure 4A*). We plotted protein intensity values against mRNA expression values (normalized protein intensity vs. mRNA transcripts per million (TPM)) and calculated Spearman correlation coefficients ($\rho$) to determine the degree of monotonic relationship between mRNA and protein for HSCs, MPPas, MPPbs and MPPcs. The correlation was lowest in the HSC compartment ($\rho = 0.300$), with comparable levels between MPPs (*Figure 4B and C* and *Figure 4—figure supplement 1*). Importantly, these correlation values are similar to what has been previously reported for a mixed population of human HSCs and MPPs (*Amon et al., 2019*). Pearson correlation of genes that were detected as mRNA and protein in all four cell types revealed largest difference in fold changes (normalized protein intensity/mRNA TPM) when comparing HSC to MPP values, but less so between

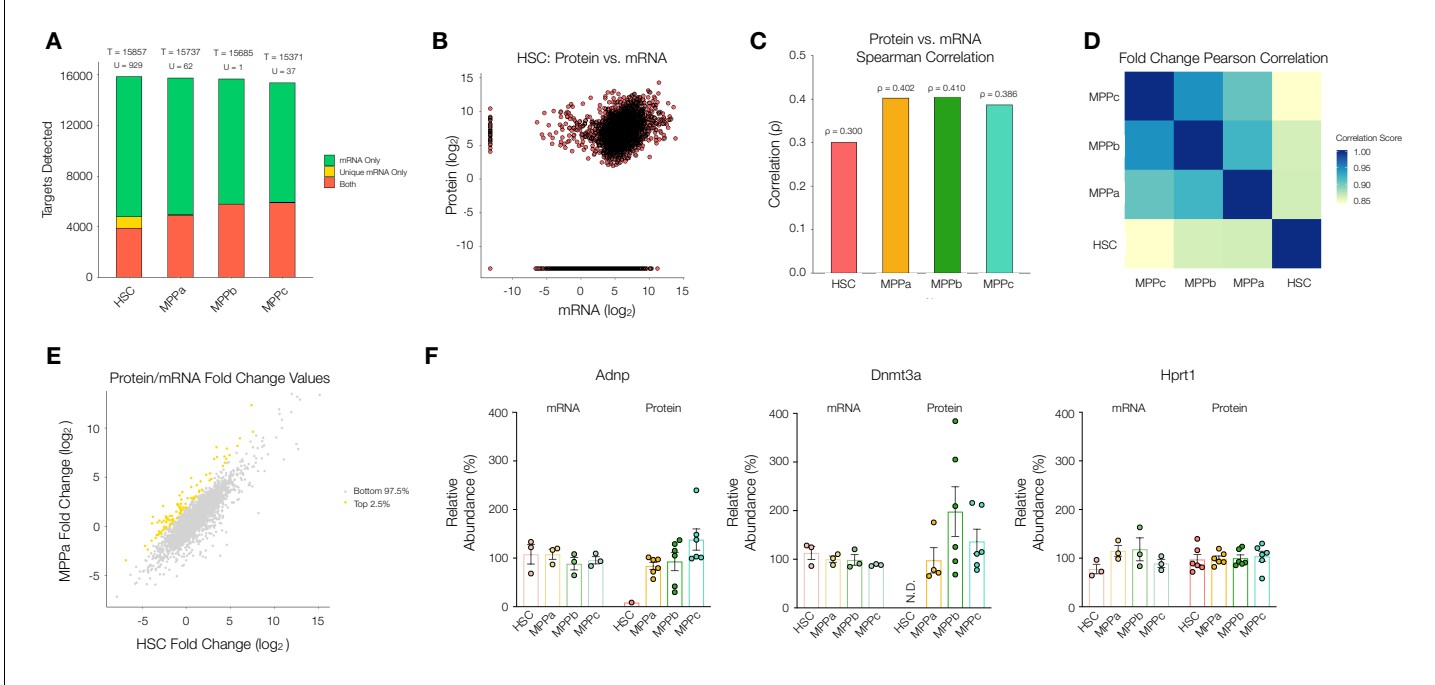

**Figure 4.** Comparison between the proteome and transcriptome of HSCs and MPPs. (**A**). Within the transcriptome, count of genes detected as mRNA only (Green), mRNA only uniquely to a given cell type (Yellow), or both protein and mRNA (Orange). T = total count of genes detected across proteome and transcriptome (sum of all bars) per cell type. U = mRNA only uniquely to a given cell type (yellow bar). (**B**) Log$_2$ normalized protein intensity vs Log$_2$ mRNA TPM for all genes detected in young adult mouse HSCs. 0.0001 was added to normalized data to account for zeroes. (**C**) Protein vs mRNA Spearman correlation value for each cell type. (**D**) Pearson correlations between combinations of HSC and MPPs for Log$_2$ normalized protein intensity/mRNA TPM fold-change values of genes detected in proteome and transcriptome across all four cell types. (**E**) Log$_2$ normalized protein intensity/mRNA TPM fold-change values of HSC vs MPPa for genes detected in proteome and transcriptome of both cell types. Top 2.5% genes with highest MPP fold-change/HSC fold-change ratios (Yellow), identifying genes where there is reduced protein per mRNA in the HSC compartment compared to MPPs. (**F**) Relative mRNA TPM value and protein intensity value of the genes Adnp, Dnmt3a and Hprt1 (housekeeping gene) across HSC and MPPs. To determine the relative values, average intensity and average TPM was calculated across all experimental replicates across all cell types, for MS and RNA-sequencing, respectively. The percentage with respect to the average was calculated and graphed for each replicate. Error bars represent standard error to the mean. For B-E, proteomic replicates were averaged across non-zero values. Transcriptome values were averaged across all values. N.D. = not detected in any replicate; TPM = transcripts per million.

The online version of this article includes the following figure supplement(s) for figure 4:

**Figure supplement 1.** Protein detection *vs.* mRNA expression (log2) for MPPs.

**Figure supplement 2.** Principal component analysis (PCA) of Protein/mRNA fold-change for each cell type.

**Figure supplement 3.** Log$_2$-fold change in Protein vs. mRNA values for each cell type for proteins detected in all 4 cell types MPPb vs. HSC and MPPc vs. HSC.

**Figure supplement 4.** Null distribution analysis to validate significance.

MPPs (*Figure 4D*). In fact, such a difference was observed on PCA of fold changes of genes with cell types as features. With the original bases of cell type features projected onto the PCA, HSC distinctly points in opposite direction along PC2 (*Figure 4—figure supplement 2*). These analyses support previous reports by Signer and co-workers that mRNA translation is uniquely regulated in the HSC compartment at least in part through a mechanism other than altered gene transcription (*Hidalgo San Jose et al., 2020*; *Signer et al., 2014*). Given these distinct differences observed between HSCs and MPPs, and the potential for post-transcriptional regulation of protein abundance, we searched for proteins that were uniquely low in HSCs or uniquely not present in HSCs, but highly detected or present as mRNA (*Figure 4A and E* and *Figure 4—figure supplement 3*). To detect uniquely low proteins in HSCs, we plotted normalized protein intensity/mRNA TPM fold changes of HSCs against every MPP for all genes and selected the top 2.5% of genes with differentially higher fold changes in MPPs compared to HSCs (i.e. more protein detected per mRNA in MPPs compared to HSCs). To identify uniquely undetected proteins, we further segmented all genes in the

transcriptome for each cell type by genes that were detected by RNA-seq only (mRNA Only, green bars), RNA-seq only unique to a cell type (Unique mRNA Only (U), yellow bars), and detected by both RNA-seq and MS analysis (Both, orange bars) (*Figure 4A*). In order to be considered a uniquely undetected protein for a cell type, the protein was required to be detected by MS in at least 3 replicates of another cell type to ensure that these proteins are readily detectable by MS analysis and therefore confidently absent in the cell type of interest. We observed that of the proteins that were not detected in HSCs despite the presence of mRNA, a large number of them were unique to HSCs—more so than by random chance compared to MPPs (*Figure 4A* and *Figure 4—figure supplement 4*). This suggests that the lower diversity of proteins in HSCs is likely attributed to biological, rather than technical, reasons.

This list of genes where message and protein abundance are decoupled uniquely in the HSC compartment include Activity-dependent neuroprotector homeobox (Adnp) and Dnmt3a, among others (*Figure 4F*, *Supplementary file 1*, Table 9). Adnp is a transcriptional regulator implicated in neural development that also affects erythropoiesis (*Dresner et al., 2012*; *Mandel et al., 2007*). While mRNA transcripts of Dnmt3a and Adnp were detected in the HSC compartment at comparable levels to that of MPPs, protein levels were markedly reduced for Adnp and absent for Dnmt3a in HSCs (*Figure 4F*). However, the mRNA and protein levels of our housekeeping protein Hprt1 were consistent across all cell types (*Figure 4F*). Uniquely decoupled mRNA and protein levels of the biologically relevant protein Dnmt3a supports the hypothesis that HSCs have distinct regulatory mechanisms downstream of transcription that play a role in HSC biology.

Given reduced correlation between mRNA and protein abundance in the HSC compartment, we wondered if this difference could be attributed to protein translation and/or protein degradation. Previous literature reports highlight a marked decrease in rates of translation (*Jarzebowski et al., 2018*; *Signer et al., 2014*). Enrichment analysis of our mass spectrometry data also reveals a reduction in abundance of proteins associated with ribosome biogenesis in the HSC compartment (*Figure 5A*). Consistent with these findings, we measured cellular levels of total RNA in each cell type. Comparable to previous reports, we determined total RNA content in HSCs to be approximately 1 pg/cell and a significant increase in total RNA content in all progenitor populations in comparison after normalization with respect to cell size (*Figure 5B*, *Signer et al., 2014*; *Jarzebowski et al., 2018*). By mass spectrometry we detected lower levels of ribosomal proteins in young adult mouse HSCs (*Figure 5—figure supplement 1*). Taken together, these data further support a scenario where rates of translation and potentially lower ribosome component abundance are in part responsible for regulating reduced protein abundance in the young adult HSC compartment.

## Potential regulatory mechanisms for discordance between protein and mRNA levels

It has been reported previously that miRNA expression plays a critical role in HSC maintenance, expansion and, in downstream progenitors, fate commitment (*Chung et al., 2011*). Given the significant reduction in protein diversity in HSCs despite a similar number of genes detected at the transcript level, we wondered if miRNAs were contributing to this reduction in protein abundance. Across the transcriptome of HSCs and MPPs ~ 86% of genes are potential targets of miRNAs for all cell types according to the miRDB database (*Figure 5C*). Within the list of genes that are putative targets of miRNA, genes that are uniquely detected at the mRNA level but absent at the protein level are more enriched in the HSC dataset than by random chance (881 in total) (*Figure 5C* and *Figure 5—figure supplement 2*). The mRNA expression values of these unique genes in the HSC compartment do not deviate strongly compared to mRNA expression values in MPPs, and therefore transcriptional regulation cannot sufficiently explain the absence of their low detection by mass spectrometry (*Figure 5—figure supplement 3*). To consider which miRNAs may be most responsible for the lower proteome diversity in young adult mouse HSCs, we counted the number of genes uniquely undetected in the proteome of HSCs that overlapped with the putative target list for each miRNA. Within the fourth quartile of miRNAs with the highest overlaps, we saw known miRNAs implicated in mouse HSC and early hematopoiesis biology, such as *Mir29a*, *Mir25a*, *Mir125b* and *Mir130a* (*Figure 5D* and *Supplementary file 1*, Table 10; *Bissels et al., 2012*; *Chung et al., 2011*; *Guo et al., 2010*; *Guo and Scadden, 2010*; *Hu et al., 2015*; *O'Connell et al., 2010*; *Ooi et al., 2010*). Notably, we have previously shown *Mir29a* to be highly expressed in HSCs compared to progenitor cells, and it has been implicated in negatively regulating Dnmt3a levels, in turn, promoting

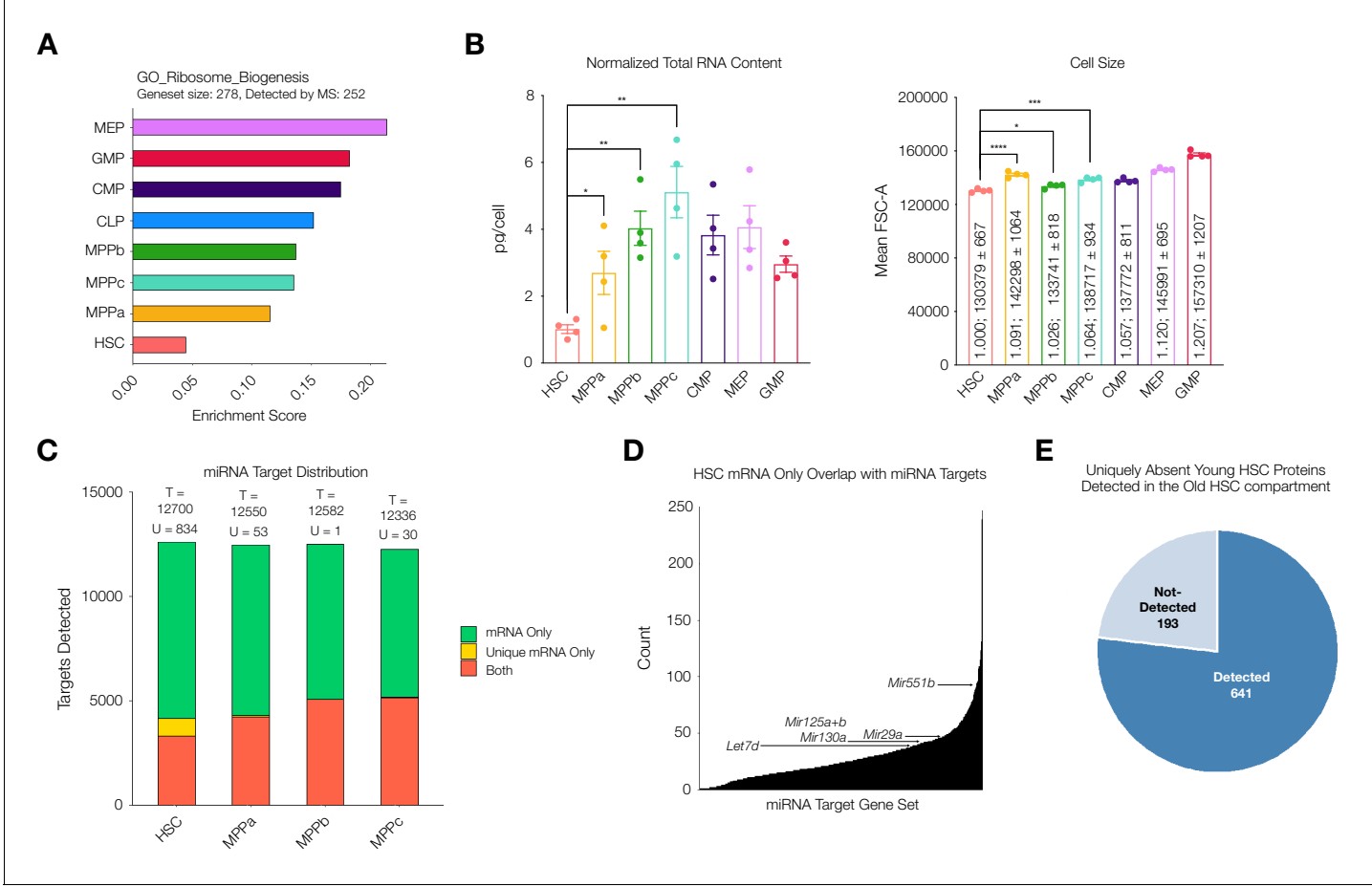

**Figure 5.** Potential mechanisms of regulation responsible for uniquely discordant protein to mRNA relationship in young adult mouse HSCs. (A) ssGSEA of proteins associated with GO Ribosome Biogenesis. P-adj = 0.00003. Enrichment scores were averaged across replicates for each cell type. FDR = 0.05 (B) Total RNA content normalized with respect to cell size in each cell type. N = 4 from 5 pooled mice (3 males, 2 females). P-values: HSC vs. MPPa * = 0.0432, HSC vs. MPPb ** = 0.0012, HSC vs MPPc ** = 0.0019. Forward Scatter Area (FSC-A) for each cell type used for normalization. Relative size and mean ± standard error to the mean (SEM) FSC-A values are denoted in the bars. N = 4 mice (2 male, 2 female). P-values: HSC vs. MPPa **** < 0.0001, HSC vs MPPb * = 0.0190, HSC vs. MPPc *** = 0.0003 Error bars represent SEM. (C) Within putative miRNA targets, count of genes detected as mRNA only (Green), mRNA only uniquely to a given cell type (Yellow), or both protein and mRNA (Orange). T = total count of putative miRNA targets. U = mRNA only uniquely to a given cell type (yellow bar). (D) Count of genes that are uniquely mRNA only for young adult mouse HSCs within a given miRNA's putative target list. Examples of potential or previously-implicated miRNAs are denoted. (E) Proteomic detection profile in old adult HSC of putative targets of miRNAs that are uniquely mRNA only in young adult mouse HSCs.

The online version of this article includes the following figure supplement(s) for figure 5:

**Figure supplement 1.** Ribosomal proteins that are uniquely very low or absent in each cell type.

**Figure supplement 2.** miRNA target null distribution analysis.

**Figure supplement 3.** Comparison of mRNA levels of miRNA protein targets that are uniquely missing in the young adult HSC compartment to MPPs reveal comparable mRNA levels between cells types.

**Figure supplement 4.** Percent of genes for each miRNA target list uniquely expressed as mRNA only in young adult mouse HSCs detected as protein in old adult mouse HSCs.

**Figure supplement 5.** ssGSEA of proteins associated with GO Epigenetic Regulation, and Ribosome Biogenesis including young and old adult mouse HSCs and progenitors.

**Figure supplement 6.** ssGSEA of proteins associated with GO Protein Monoubiquitination.

self-renewal (*Han et al., 2010*; *Hu et al., 2015*). Deletion of *Mir29a* has been shown to decrease self-renewal and increase HSC cycling (*Hu et al., 2015*). We also identified *Mir551*. While *Mir551* expression has not been validated in mouse HSCs, it has been shown to be expressed in human HSCs and MPPs and is a negative prognostic indicator in acute myeloid leukemia (*de Leeuw et al., 2016*).

Finally, we further investigated the increase of proteomic diversity detectable in old mouse HSCs compared to young adult mouse HSCs in relation to miRNAs. Of the 881 uniquely undetected young adult HSC proteins that are putative targets of miRNAs, 776 (88%) are detected in old mouse HSCs, with 105 still undetected (*Figure 5E*). With this increase in protein diversity, many putative miRNA target genes uniquely undetected at the protein level in young adult mouse HSCs are detected in old mouse HSCs (*Figure 5—figure supplement 4*). This rescue of protein diversity in old mouse HSCs may be attributed to alternative regulatory mechanisms in protein abundance between young and old adult mouse HSCs (*Figure 3A*). Notably, ribosomal proteins were more readily detected in the old HSC compartment compared to the young adult compartment (*Figure 5—figure supplement 1*). Enrichment analysis of Gene Ontology genesets reveals the lowest enrichment of proteins associated with epigenetic regulation of gene expression and ribosome biogenesis in young adult mouse HSCs, with levels in old mouse HSCs comparable to those of progenitors (*Figure 5—figure supplement 5*). This suggests a model in which the regulatory mechanism of old mouse HSCs for protein abundance is more similar to MPPs, with increased reliance on epigenetic regulation of transcription and perhaps increased translational capacity, although this has not yet been well studied. In addition to revealing the potential implications of known miRNAs on protein levels, our analysis lays the foundation for the potential discovery of novel miRNAs that play a role in HSC biology, such as *Mir551*. It also suggests potential differences in post-transcriptional regulation of the stem cell compartment in the aging process, highlighting the importance of coupling transcriptomic studies with proteomic studies to fully understand the biology of rare cell types in any system.

## Discussion

This manuscript provides deep proteomic coverage of mouse young and old adult HSCs and their progenitors and complements currently available mass spectrometry datasets (*Cabezas-Wallscheid et al., 2014*; *Jassinskaja et al., 2017*). We utilized shotgun mass spectrometry to allow for an unbiased, exhaustive characterization of the early young and old adult hematopoietic proteome to levels not yet studied, 12 cell types in all. We validated the quality of our data via multiple methods. These data are consistent with the detection of established surface markers that functionally separate HSCs and progenitors (*Figure 1E* and *Figure 1—figure supplement 4*). PCA visualizes the clustering of cell types with biological consistency across each principal component. We also validate our intensity readouts via FACS analysis and microscopy of select genes, including Esam, Pfkl, and Dnmt3a. Many of the proteomic profiles identified validate functional and qualitative studies reported by other groups, such as DNA damage pathways, and abundance of Ki67, Igf2bp2, and Hmga2. The increased detection of CD150/Slamf1 and vWF in the old HSC and MPPa compartment is consistent with previous observations of increased myeloid bias in old mouse HSCs (*Beerman et al., 2010*; *Grover et al., 2016*; *Sanjuan-Pla et al., 2013*). Curiously in the old HSC compartment, we see an increase in protein diversity compared to young adult mouse HSCs (*Figure 3A*). Additional studies are underway in our laboratory in order to understand the mechanism by which this occurs and the biochemical consequences of increased protein diversity in the stem cell compartment.

The lower proteomic diversity in HSCs compared to progenitors corroborates decreased rates of protein synthesis in the HSC compartment and stands in contrast to equally-diverse transcriptomes for HSCs and MPPs (*Ramos et al., 2006*). To this effect, some RNAs may not be translated, and their presence could reflect the opening of their chromatin rather than the need for these proteins to be translated within the HSC compartment. Correlation studies and PCA of genes suggest differential regulation between mRNA and protein levels when comparing HSCs to their progenitors (*Figure 4C and D* and *Figure 4—figure supplement 2*). PCA of the cell types also supports the hypothesis that HSCs exhibit more regulation following gene transcription than previously appreciated, as the top 250 proteins most antagonistic to component 1 enriched for proteins involved in 'chromatin silencing', 'histone modification', and 'mRNA processing' (*Figure 1D* and *Supplementary file 1*, Table 3). Additionally, HSCs have very low levels of proteins associated with Epigenetic Gene Expression as determined with enrichment analysis (*Figure 5—figure supplement 5*). Total RNA levels are also reduced in HSCs compared to progenitor cells (*Figure 5B*), and we report a lower level of proteins associated with ribosome biogenesis in HSCs (*Figure 5A*). Previous single-cell mRNA quantification studies in E-SLAM HSCs (EPCR$^+$, CD48$^-$, CD150$^+$), LMPP (Lin$^-$, cKit$^+$, Sca1$^+$, Flk2$^+$, CD34+$^+$), GMP

(Lin⁻, cKit⁺, Sca1⁺, Flk2⁺, CD16/32⁺, CD34⁺) and MEPs (Lin⁻, cKit⁺, Sca1⁺, CD16/32⁻, CD34⁻) revealed a steady increase in total mRNA during differentiation, with E-SLAM HSCs having the lowest levels of total mRNA (*Nestorowa et al., 2016*). While these experiments were performed on cells sorted using very different gating strategies compared to our own, our reports do not stand in contrast these findings. It is possible that HSCs, while having less mRNA content and reduced transcription, still maintain higher diversity in message transcribed. Taken together with previous literature reports on translation and chromatin structure in the HSC compartment, these data suggest a new hypothesis in the regulation of stem maintenance and HSC homeostasis, wherein HSCs undergo a loss of diversity from gene accessibility to protein (*Figure 6*).

It has been reported that HSCs have more open chromatin than MPPs, suggesting an increased plasticity in gene transcription (*Buenrostro et al., 2018*). Conversely, mRNA translation is markedly reduced and highly sensitive to perturbations in the HSC compartment, and we now report a lower correlation between mRNA and protein (*Signer et al., 2014*). We therefore propose that perhaps primary regulatory mechanisms of stemness shift downstream of gene transcription towards translation specifically in the young adult HSC compartment (*Figure 6*). These results are further supported by previous studies in a mixed population of human HSCs and MPPs where the correlation between mRNA and protein, particularly for genes critical for stem-cell maintenance, such as quiescence and telomere maintenance, was markedly reduced compared to more- committed progenitor populations (*Amon et al., 2019*).

There are two possible contributing factors to reduced protein abundance in the HSC compartment: Rates of translation and/or rates of degradation. Our data and others suggest that rates of translation are extraordinarily critical for stem cell proteostasis (*Figure 6*, *Signer et al., 2014*). However, this does not rule out a more robust presence of protein turnover machinery mediated through proteasomal or lysosomal degradation in HSCs. Signer and co-workers recently reported the reduction of ubiquitylated proteins and misfolded proteins in HSCs and that these levels are sensitive to increased rates of translation (*Hidalgo San Jose et al., 2020*). They also demonstrate that increasing levels of misfolded proteins can result in proteasomal stress and elicit the unfolded protein response. Our dataset reveals that proteins associated with the monoubiquitination are less abundant in HSCs (*Figure 5—figure supplement 6*). While these data indicate that proteasomal degradation is not a major contributing factor to reduced protein levels, at least in young adult mouse HSCs, they do not take into account the possibility of lysosomal degradation, which has previously been demonstrated to be important in neural stem cell maintenance and quiescence (*Leeman et al., 2018*). While we

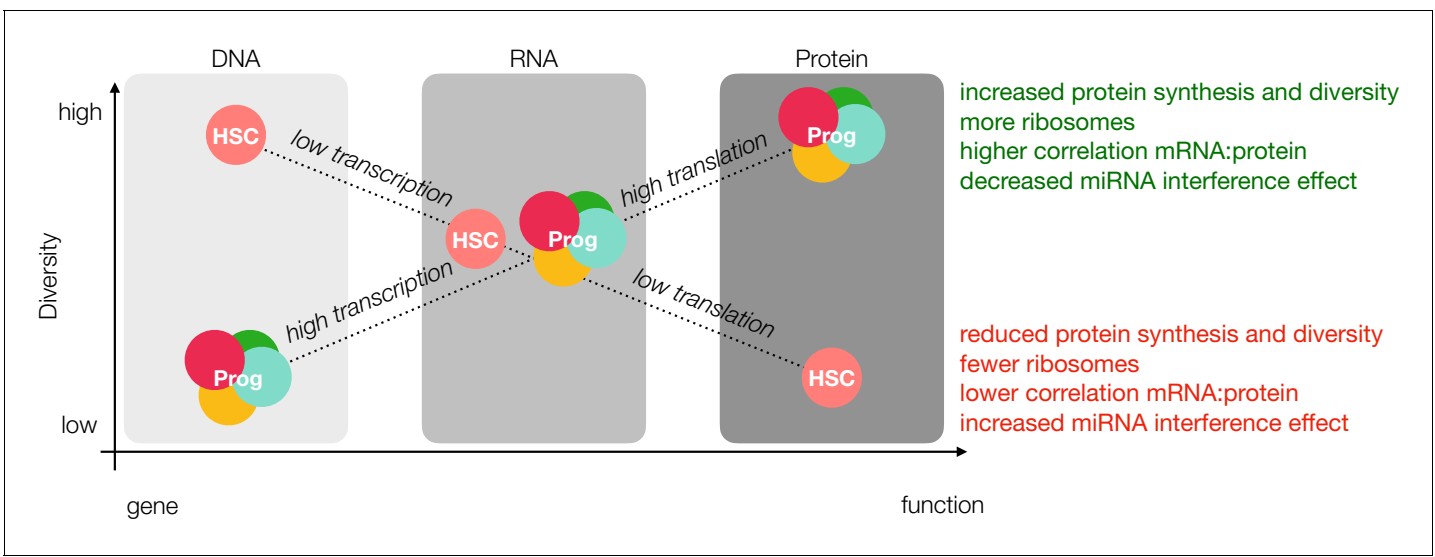

**Figure 6.** Loss of diversity hypothesis. In our model, young adult mouse HSCs have more open chromatin than progenitor cells; however, lower rates of transcription result in comparable levels of message diversity at the mRNA level. Decreased rates of translation due to fewer ribosomes and reduced ribosomal activity, miRNA interference and sensitivity towards the unfolded protein response result in less protein diversity as detected by mass spectrometry. Prog = Progenitor.

report here the findings of our mass spectrometry characterization of early hematopoiesis, including this unique discordance between mRNA and protein in the young adult HSC compartment, there is still much to uncover from a mechanistic perspective. Future studies in our lab are underway to identify the mechanisms which result in decreased proteome diversity in young adult HSC and whether such a phenomenon persists during aging and in development.

These data provide a deeper understanding of proteins expressed during young and old adult hematopoiesis that are currently detectable by mass spectrometry. However, we caution that this resource is by no means a complete list of proteins expressed by every cell in early hematopoiesis. As mass spectrometry methods continue to allow for improved data coverage with low amounts of protein and characterization of hematopoietic cells becomes more nuanced, deeper proteomic characterizations of hematopoiesis will become possible. It also will open up the opportunity to further segment stem and progenitor cell fractions, such as the fractionation of the HSC compartment based on the different stages of cell cycle. This can be important for our analysis of protein changes throughout early hematopoiesis, as HSCs are more quiescent than downstream progenitors, which can contribute to differential abundance of proteins as a consequence of cell cycle. This is highlighted by our analysis of Dnmt3a, which is not expressed until quiescent stem cells enter cycle, wherein the daughter cells could include MPP or OPP, necessitating closing expression of some genes operative in stem cells but not MPP. In addition, Signer, Morrison and co-workers have reported that differences in the rates of translation between HSCs and MPPs cannot be entirely explained by cell cycle (*Signer et al., 2014*). Our data reveal global differential regulation in protein abundance, some of which will be cell cycle-dependent as well as cycle-independent. Given the dearth of proteomic information currently available for early hematopoietic cells in young and old adult mice, these data reveal previously-uncharacterized suites of proteins detectable in young and old HSCs and progenitors. The nature of these expansive data allows not only for the identification of novel surface markers of each cell type but also a deeper understanding of intracellular regulatory proteins of transcription and translation that contribute to stem- and progenitor-cell quiescence survival, fate commitment and function.

# Materials and methods

## Key resources table

| Reagent type (species) or resource | Designation | Source or reference | Identifiers | Additional information |
|---|---|---|---|---|
| Antibody | Rat monoclonal anti-mouse CD34 (RAM34) FITC | ThermoFisher Scientific | Cat# 11-0341-82, RRID:AB_465021 | FC (5 ug/mL) |
| Antibody | Rat monoclonal anti-mouse Lineage cocktail A700 (anti-mouse CD3, clone 17A2; anti-mouse Ly-6G/Ly-6C, clone RB6-8C5; anti-mouse CD11b, clone M1/70; anti-mouse CD45R/B220, clone RA3-6B2; anti-mouse TER-119/Erythroid cells, clone Ter-119) | Bio-Legend | Cat# 133313, RRID:AB_2715571 | FC (5 uL/mouse) |
| Antibody | Rat monoclonal anti-mouse cKIT (2B8) APC-eFluor780 | ThermoFisher Scientific | Cat# 47-1171-82, RRID:AB_1272177 | FC (2 ug/mL) |
| Antibody | Rat monoclonal anti-mouse Sca1 (D7) PE-Cy7 | Bio-Legend | Cat# 108114, RRID:AB_493596 | FC (2 ug/mL) |
| Antibody | Rat monoclonal anti-mouse CD150 (TC15-12F12.2) APC | Bio-Legend | Cat# 115910, RRID:AB_493460 | FC (2 ug/mL) |
| Antibody | Rat monoclonal anti-mouse Flt3 (A2F10) PerCP-eFluor710 | ThermoFisher Scientific | Cat# 46-1351-82, RRID:AB_10733393 | FC (2 ug/mL) |
| Antibody | Rat monoclonal anti-mouse CD16/32 (2.4G2) BUV395 | BD Biosciences | Cat# 740217, RRID:AB_2739965 | FC (2 ug/mL) |
| Antibody | Rat monoclonal anti-mouse IL7Ra (A7R34) APC | Bio-Legend | Cat# 135012, RRID:AB_1937216 | FC (2 ug/mL) |

*Continued on next page*

*Continued*

| Reagent type (species) or resource | Designation | Source or reference | Identifiers | Additional information |
|---|---|---|---|---|
| Antibody | Rat monoclonal anti-mouse CD150 (TC15-12F12.2) BV421 | Bio-Legend | Cat# 115925, RRID:AB_10896787 | FC (2 ug/mL) |
| Antibody | Rat monoclonal anti-mouse IL7Ra (SB/199) BV711 | BD Biosciences | Cat# 565490, RRID:AB_2732059 | FC (2 ug/mL) |
| Antibody | Rat monoclonal anti-mouse ESAM (1G8) APC | Bio-Legend | Cat# 136207, RRID:AB_2101658 | FC (2 ug/mL) |
| Antibody | Rat monoclonal anti-mouse CD16/32 (93) PE | Bio-Legend | Cat# 101307, RRID:AB_312806 | FC (2 ug/mL) |
| Antibody | Rabbit monoclonal anti-mouse Pfkl (EPR11904) | Abcam | Cat# ab181064, RRID | IF (1:100) |
| Antibody | Cy3 AffiniPure F(ab')2 Fragment Donkey Anti-Rabbit IgG | Jackson ImmunoResearch | Cat# 711-166-152, RRID:AB_2313568 | IF (1:500) |
| Commercial assay or kit | iST NHS 96x | PreOmics | iSTNHS96x | |
| Commercial assay or kit | Pierce Quantitative Colorimetric Peptide Assay | ThermoFisher Scientific | 23275 | |
| Commercial assay or kit | RNAeasy minelute CleanUp Kit | QIAGEN | 74204 | |
| Commercial assay or kit | NEBNext Ultra DNA Library Prep Kit for Illumina | New England BioLabs | E7103 | |
| Other | TRIzol | Invitrogen | 15596018 | |
| Other | RQ1 RNase free DNase | Promega | M6101 | |
| Other | Agencourt Ampure XP | Beckman Coulter | A63881 | |

## Lead contact and materials availability

Further information and requests for resources should be directed to and will be fulfilled by the Corresponding Author, Irving Weissman (irv@stanford.edu). This study did not generate new reagents. The mass spectrometry proteomics data have been deposited to the ProteomeXchange Consortium via the PRIDE partner repository with the dataset identifier PXD017442 and 10.6019/PXD017442.

## Experimental model details
### Animals

An in-house C57BL/6 strain of mice was used for collection of bone marrow-derived young adult mouse HSCs and progenitors at 8–14 weeks of age. For old adult mouse studies, C57BL/6 mice (24–27 months) were a gift from Charles Chan. An equal number of male and female mice were used across all experiments. For young adult studies, 50 mice were used for each biological cohort. For old adult studies, two mice were used for each biological cohort. Cells were pooled for each cohort across multiple sorts with sort numbers documented for mass spectrometry sample preparation. Care was taken to ensure that each cell type from a biological cohort within a processed group of cells (HSCs and MPPs together; CMPs, GMPs, MEPs together; CLPs together), had equal contribution from each sort within the cohort. For example, given the rarity of HSCs, every HSC that could be purified during each sort was sorted, but for the more abundant MPPcs, 100,000 cell aliquots were sorted and lysis scaled accordingly. Mice for all experiments were immunocompetent and group housed in an AAALAC certified barrier facility. The light cycle in the facility is 12 hr on/12 hr off. All experiments were performed according to guidelines established by the Stanford University Administrative Panel on Laboratory Animal Care under protocol #10266.

## Method details
### Data collection and processing

In all downstream cell processing, FACS buffer (2% fetal bovine serum (FBS, US Origin, HyClone, Cytiva, Marlborough, Massachusetts) in phosphate buffered saline (PBS, pH 7.4, calcium and

magnesium free, Gibco, ThermoFisher Scientific, Waltham, Massachusetts)) was used at ice-cold temperature unless otherwise stated. All cell pelleting was done at 1,300 rpm for 5 min at 4°C unless otherwise stated.

## Isolation of mouse bone marrow cells

Mice were euthanized and hips, femurs, tibia, humeri and spine harvested. Bones were cleaned and crushed with a mortar and pestle to retrieve resident bone marrow cells with FACS buffer. Cells in FACS buffer were passed through at 40 µm filter and pelleted.

## Isolation and purification of mouse HSCs and MPPs

Filtered and pelleted marrow cells were resuspended in 800 µL FACS buffer per mouse. Miltenyi cKit enrichment beads were added (15 µL per mouse, Miltenyi Biotec, Sunnyvale, California) and incubated for 15 min at 4°C. After incubation, cells were washed with 10 mL FACS buffer per mouse and pelleted. Cell samples were resuspended in 1 mL FACS buffer per mouse and loaded on to a Miltenyi MACS magnetic separation column (Miltenyi Biotec, Sunnyvale, California). The column was washed with 3 mL FACS buffer three times. cKit$^+$ cells were eluted according to manufacturer's protocol and pelleted. Enriched cells were resuspended in FACS buffer (100 µL per mouse) and Anti-CD34 FITC (clone RAM34, ThermoFisher Scientific, Waltham, Massachusetts; 5.0 µg/mL final concentration) was added. Cells were incubated for 45 min on ice prior to addition of the remaining antibodies: Anti-Lineage Cocktail A700 (Biolegend, San Diego, California; 5 µL per mouse), Anti-cKIT APC-eFluor780 (clone 2B8, ThermoFisher Scientific, Waltham, Massachusetts; 2 µg/mL final concentration), Anti-Sca1 PE-Cy7 (clone D7, Biolegend, San Diego, California; 2 µg/mL final concentration), Anti-CD150 APC (clone TC15-12F12.2, Biolegend, San Diego, California; 2 µg/mL final concentration), and Anti-Flt3 PerCP-eFluor710 (clone A2F10, Biolegend, San Diego, California; 2 µg/mL final concentration). The cells were incubated on ice for an additional 30 min with the complete cocktail prior to dilution with FACS buffer (10 mL) to remove excess antibody. Cell were pelleted and resuspended in 1 mL/ mouse fresh FACS buffer containing SYTOX Blue (ThermoFisher Scientific, Waltham, Massachuetts; 1:3000) prior to sorting on a BD FACS Aria (BD Biosciences, San Jose, California).

## Isolation and purification of mouse CMP, GMP, MEPs

Filtered and pelleted marrow cells were resuspended in Gibco ACK lysis buffer (ThermoFisher Scientific, Waltham, Masschusetts; 1 mL) and incubated for 5 min at ambient temperature. Lysis was then quenched with 10 mL of FACS buffer, and cells pelleted. RBC-depleted cells were resuspended in 800 µL FACS buffer per mouse. Miltenyi Lineage depletion beads were added (Miltenyi Biotec, Sunnyvale, California; 100 µL per mouse) and incubated for 10 min at 4°C. After incubation, cells were loaded on to a Miltenyi MACS magnetic separation column. The column was washed with 3 mL FACS buffer three times. The flow-through with Lin$^-$ cells was pelleted and resuspended in FACS buffer (100 µL per mouse). Anti-CD34 FITC (clone RAM34; 5.0 µg/mL final concentration) was added. Cells were incubated for 45 min on ice prior to addition of the remaining antibodies: Anti-Lineage Cocktail A700 (3 µL per mouse), Anti-cKIT APC-eFluor780 (clone 2B8; 2 µg/mL final concentration), Anti-Sca1 PE-Cy7 (clone D7; 2 µg/mL final concentration), Anti-CD150 APC (clone TC15-12F12.2; 2 µg/mL final concentration), and Anti-CD16/32 BV395 (BD Biosciences, San Jose, California; clone 2.4G2; 2 µg/mL final concentration). The cells were incubated for an additional 30 min on ice with the complete cocktail prior to washing with FACS buffer to remove excess antibody (10 mL). Cell were pelleted and resuspended in 500 µL/mouse fresh FACS buffer containing SYTOX Blue (1:3000) prior to sorting on a BD FACS Aria.

## Isolation and purification of mouse CLPs

Lineage depletion protocol up to antibody staining was identical to isolation and purification of mouse CMP, GMP and MEPs. After resuspending in FACS buffer (100 µL per mouse), Cells were incubated on ice for 30 min with the following antibodies: Anti-Lineage Cocktail A700 (3 µL per mouse), Anti-cKIT APC-eFluor780 (clone 2B8; 2 µg/mL final concentration), Anti-Sca1 PE-Cy7 (clone D7; 2 µg/mL final concentration), Anti-Flt3 PerCP-eFluor710 (clone A2F10; 2 µg/mL final concentration), and Anti-IL7Ra APC (clone A7R34, BD Biosciences, San Jose, California; 2 µg/mL final

concentration). Upon incubation completion, cells were diluted with FACS buffer (10 mL) to remove excess antibody. Cell were pelleted (1,300 rpm x 5 min, 4°C) and resuspended in 500 μL/mouse fresh FACS buffer containing SYTOX Blue (1:3000) prior to FACS. Samples were sorted on a BD FACS Aria.

## Isolation and purification of mouse LSK cells

Filtered and pelleted marrow cells were resuspended in 800 μL FACS buffer per mouse. Miltenyi cKit enrichment beads were added (15 μL per mouse) and incubated for 15 min at 4°C. After incubation, cells were washed with 10 mL FACS buffer per mouse and pelleted. Cell samples were resuspended in 1 mL FACS buffer per mouse and loaded on to a Miltenyi MACS magnetic separation column. The column was washed with 3 mL FACS buffer three times. cKit$^+$ cells were eluted according to manufacturer's protocol and pelleted. Enriched cells were resuspended in FACS buffer (100 μL per mouse). Cells were incubated on ice with the following antibodies: Anti-Lineage Cocktail A700 (3 μL per mouse), Anti-cKIT APC-eFluor780 (clone 2B8; 2 μg/mL final concentration), Anti-Sca1 PE-Cy7 (clone D7; 2 μg/mL final concentration). Upon incubation completion, cells were diluted with FACS buffer (10 mL) to remove excess antibody. Cell were pelleted (1,300 rpm x 5 min, 4°C) and resuspended in 1 mL/mouse fresh FACS buffer containing SYTOX Blue (1:3000) prior to FACS. Samples were sorted on a BD FACS Aria.

## FACS analysis – Esam expression validation

Each mouse was processed as an individual biological replicate (N = 5, three male, two female). Lineage depletion protocol up to antibody staining was identical to isolation and purification of mouse CMP, GMP and MEPs. After resuspending in FACS buffer (100 μL per sample), Anti-CD34 FITC (clone RAM34; 5.0 μg/mL final concentration) was added. Cells were incubated for 45 min on ice prior to addition of the remaining antibodies: Anti-Lineage Cocktail A700 (3 μL per mouse), Anti-cKIT APC-eFluor780 (clone 2B8; 2 μg/mL final concentration), Anti-Sca1 PE-Cy7 (clone D7; 2 μg/mL final concentration), Anti-CD150 BV421 (clone TC15-12F12.2; 2 μg/mL final concentration), Anti-Flt3 PerCP-eFluor710 (clone A2F10; 2 μg/mL final concentration), Anti-IL7Ra BV711 (clone SB/199, BD Biosciences, San Jose, California; 2 μg/mL final concentration), Anti-CD16/32 PE (clone 93, Biolegend, San Diego, California; 2 μg/mL final concentration), and Anti-ESAM APC (clone 1G8, Biolegend, San Diego, California; 2 μg/mL final concentration). The cells were incubated for an additional 30 min on ice with the complete cocktail prior to washing with FACS buffer (1 mL) to remove excess antibody. Cells were pelleted and resuspended in 250 μL fresh FACS buffer containing DAPI (ThermoFisher Scientific, Waltham, Massachusetts; 1:1000 stock solution) prior to analysis.

## FACS analysis

Each mouse was processed as an individual biological replicate (N = 5, three male, two female). Lineage depletion protocol up to antibody staining was identical to isolation and purification of mouse CMP, GMP and MEPs. After resuspending in FACS buffer (100 μL per sample), Anti-CD34 FITC (clone RAM34; 5.0 μg/mL final concentration) was added. Cells were incubated for 45 min on ice prior to addition of the remaining antibodies: Anti-Lineage Cocktail A700 (3 μL per mouse), Anti-cKIT APC-eFluor780 (clone 2B8; 2 μg/mL final concentration), Anti-Sca1 PE-Cy7 (clone D7; 2 μg/mL final concentration), Anti-CD150 BV421 (clone TC15-12F12.2; 2 μg/mL final concentration), Anti-Flt3 PerCP-eFluor710 (clone A2F10; 2 μg/mL final concentration), Anti-IL7Ra BV711 (clone SB/199, BD Biosciences, San Jose, California; 2 μg/mL final concentration), Anti-CD16/32 PE (clone 93, Biolegend, San Diego, California; 2 μg/mL final concentration), and Anti-ESAM APC (clone 1G8, Biolegend, San Diego, California; 2 μg/mL final concentration). The cells were incubated for an additional 30 min on ice with the complete cocktail prior to washing with FACS buffer (1 mL) to remove excess antibody. Cells were pelleted and resuspended in 250 μL fresh FACS buffer containing DAPI (ThermoFisher Scientific, Waltham, Massachusetts; 1:1000 stock solution) prior to analysis. Flow cytometry data was processed using FlowJo v10.7 and the gating strategy described in *Figure 1—figure supplement 1*. Mean FSC-A was calculated using FlowJo v10.7 for each mouse and analyzed by GraphPad Prism.

## Fluorescence microscopy

For Pfkl staining, HSCs, MPPas and MPPbs were purified as described above from four mice (two male, two female) and spun in a Cytospin centrifuge onto superfrost plus glass slides (ThermoFisher Scientific, Waltham, Massachusetts) for 10 min at 1000 rpm. Upon spin completion, slides were dried for 5 min and a circle drawn around the cells with a wax pen.

For Dnmt3a and Ki67 staining, fresh cells were purified as described above and pipetted onto black epoxy-coated 21-well glass slides (Matsunami Glass Company, Bellingham, Washington). Cells for culture were first sorted into a 96-well plate with growth factors as reported previously prior to pelleting and transferring to black epoxy-coated 21-well glass slides (*Wilkinson et al., 2019*). Slides were precoated with poly-L-lysine solution (SigmaAldrich, St. Louis, Missouri;1:10 dilution of 0.01% poly-L-lysine stock solution to a final concentration of 0.001%) for 1 hr and washed twice with 50 µL PBS prior to cell addition. Cells were allowed to lie down on the slides for 15 min.

Fixation buffer (4% PFA in PBS) was added on top of the cells and incubated for 10 min. Fixative was pipetted away and the cells washed with PBS for 5 min, 3 times. Cells were incubated with permeabilization buffer (0.1% Triton X-100 in PBS) for 10 min before permeabilization buffer was removed and replaced with blocking buffer (5% donkey serum in 0.1% TritonX-100 in PBS). Cells were incubated in blocking buffer for 16 hr at 4 ℃. Primary antibody (Pfkl (EPR11904, Abcam, Cambridge, UK); Dnmt3a (64B1446, Novus Biologicals, Littleton, Colorado)); or Ki67-AF488 (D3B5, Cell Signaling Technology, Danvers, Masschusetts) in blocking buffer (1:100) was added to cells and cells were incubated for 2 hr. Upon completion, slides were washed with PBS for 5 min, three times, and appropriate secondary added (Donkey anti-mouse IgG Highly Cross-Absorbed Secondary A647, ThermoFisher Scientific, Waltham, Massachusetts or Cy3 Affinipure F(ab')2 Fragment Donkey Anti-Rabbit IgG, Jackson ImmunoResearch, West Grove, Pennsylvaniaa; 1:500 in blocking buffer). Cells were allowed to incubate with secondary for 1 hr prior to washing with PBS for 5 min, three times. DAPI (1:100, final concentration 200 ng/mL in PBS) was added to the slides and allowed to incubate for 10 min prior to mounting.

## Fluorescence image quantification

All quantification was done using ImageJ. Individual cells were manually outlined on the DAPI channel. Using the outline, integrated density and area was measured for all cells across all channels. For every channel for every cell type, mean fluorescence of the background was measured at locations with no cells. Corrected Total Cell Fluorescence (CTCF) was calculated as the following: CTCF = Integrated Density – (Area * Background Mean Fluorescence). To compare between cells, CTCF value of a protein for each cell was normalized with the CTCF value of DAPI for each cell. Significance was determined using unpaired t-test.

## Processing of purified cell types for mass spectrometry analysis

Sorted cells were pelleted and washed twice with ice-cold PBS to remove any remaining FBS (1,300 rpm x 5 min, 4℃). PBS was aspirated away, and the pellets were snap frozen with liquid nitrogen prior to storage at −80℃. Prior to lysis, cells were thawed on ice and subjected to sample preparation with the PreOmics iST NHS kit (PreOmics, Planegg, Germany) according to literature protocol. (To normalize lysis across cell number, 10 µL of lysis buffer was added for every 100,000 cells). The only additional modification made to the protocol was scaling down in volume of digest buffer to align with amount of lysis buffer (for example, 20 µL digest buffer for 20 µL lysis buffer). Samples were resuspended in 12 µL of LC-Load Buffer from the iST NHS kit and peptide concentration determined (Pierce Quantitative Colorimetric or Fluorescent Peptide Assay, ThermoFisher Scientific, Waltham, Massachusetts). Sample concentration was normalized to 100 ng/µL and 2 µL was loaded onto the instrument.

## Mass spectrometry analysis – liquid chromatography and timsTOF Pro

A nanoElute was attached in line to a timsTOF Pro equipped with a CaptiveSpray Source (Bruker, Hamburg, Germany). Chromatography was conducted at 40℃ through a 25 cm reversed-phase Aurora Series C18 column (IonOpticks, Middle Camberwell, Australia) at a constant flow-rate of 0.4 µL/min. Mobile phase A was 98/2/0.1% Water/MeCN/Formic Acid (v/v/v) and phase B was MeCN with 0.1% Formic Acid (v/v). During a 120 min method, peptides were separated by a 4-step linear

gradient (0% to 15% B over 60 min, 15% to 23% B over 30 min, 23% to 35% B over 10 min, 35% to 80% over 10 min) followed by a 10 min isocratic flush at 80% for 10 min before washing and a return to low organic conditions. Experiments were run as data-dependent acquisitions with ion mobility activated in PASEF mode. MS and MS/MS spectra were collected with $m/z$ X00 to 1500 and ions with $z = +$one were excluded.

## Mass spectrometry data analysis

Raw data files were processed with Byonic software (Protein Metrics, Cupertino, California). Fixed modifications included +113.084 C. Variable modifications included Acetyl +42.010565 N-term, pyro-Glu −17.026549 N-term Q, pyro-Glu −18.010565 N-term E. Precursor tolerance 30.0 ppm.

## Data compilation

Raw files were read for UniProtIDs, gene names and their respective mappings. 'nan', '' (empty strings), and '2 SV' were ignored. Mappings between gene names and UniProtIDs were not one-to-one. Some genes and UniProtIDs were unmapped. Therefore, before compiling the raw data, a comprehensive, one-to-one mapping was first made. Using the Retrieve/ID mapping program at www.uniprot.org (release 2019–11), all gene names were mapped to all possible UniProtIDs, and UniprotIDs were mapped to all possible gene names. The mappings from raw files and UniProt were combined to group equivalent gene names and UniProtIDs (usually with isoforms) together. From each group, one gene name and one UniProtID were selected for downstream data compilation. Any UniProtID or gene name that was not mapped was either given a protein ID (UNM #) or a gene name (Unm #). 38 gene names had no UniProtIDs and 8 UniprotIDs had no gene names. All raw files were compiled into a single data table with the selected UniProtIDs and gene names (*Supplementary file 1*, Table 1).

## RNA isolation and library preparation

RNA was isolated with TRIzol (Invitrogen, ThermoFisher Scientific, Waltham, Massachusetts) as per the manufacturer's recommendations and was further facilitated by using linear polyacrylamide as a carrier during the procedure. We treated the total RNA samples with RQ1 RNase free DNase (Promega, Madison, Wisconsin) to remove minute quantities of genomic DNA if present. DNase treated samples were cleaned using RNAeasy minelute columns (Qiagen, Hilden, Germany). 1–10 ng of total RNA was used as input for cDNA preparation and amplification using Ovation RNA-Seq System V2 (NuGEN Technologies, Redwood City, California). Amplified cDNA was sheared using Covaris S2 using the following settings: duty cycle 10%, intensity 5, cycle/burst 100, total time 5 min (Covaris, Woburn, Massachusetts). The sheared cDNA was cleaned up using Agencourt Ampure XP (Beckman Coulter Life Sciences, San Jose, California). 500 ng of sheared cDNA were used as input for library preparation using NEBNext Ultra DNA Library Prep Kit for Illumina as per manufacturer's recommendations (New England BioLabs, Ipswich, Massachusetts).

## RNA-seq and data analysis

Libraries were sequenced using NextSeq 500 (Illumina, San Diego, California) to obtain 2 × 150 base pair paired-end reads and HiSeq 2000 (Illumina, San Diego, California) to get 2 × 100 base pair paired-end reads.

## Data normalization

For quantitative comparisons, all individual replicates in both proteomics and transcriptomics were normalized to sum to 1,000,000 (*Supplementary file 1*, Table 1 for proteome, *Supplementary file 1*, Table 8 for transcriptome). When calculating $\log_2$ fold changes of subsets of data including zeros, 0.0001 was first added to clearly separate out non-detected values from lowly-detected values. Attention was paid to ensure that this smoothing did not significantly affect non-zero values. For comparisons between cell types or between protein and mRNA, proteomic replicates within a cell type were combined by taking the average of non-zero values (*Supplementary file 1*, Table 2). Transcriptomic replicates within a cell type were combined by taking the average of all values. For analyses requiring protein to mRNA comparisons, we wanted to ensure no overlaps between proteomics and transcriptomics was missed due to gene name differences. All gene names were converted to

EntrezID using the following process. Gene names were mapped to all possible EntrezID using MGI Batch Query available on www.informatics.jax.org/batch (retrieved Jan 8) (*Mouse Genome Database Group et al., 2019*). A gene name could map to multiple EntrezIDs. Among such gene names, EntrezIDs that were of other species and those that did not belong to 'old symbol', 'related synonym', 'current symbol', 'Homologene', 'synonym' and 'Genbank' were removed. Among this list, filtering for those belonging to 'current symbol' recovered most mappings. Of those that were not recovered, all that belonged to 'old symbol' were removed. 87 gene names did not map to any EntrezID. When graphing proteomic and transcriptomic data of one specific gene side-by-side for *Figure 4F*, all intensity values or TPM values across all cell types were averaged to generate an average value for MS or RNA-seq data, respectively. We then divided each replicate value by the average value and multiplied by 100 to obtain the relative % abundance. When graphing mass spectrometry and FACS data side-by-side in *Figure 1—figure supplement 5*, average MFI and average intensity were calculated across all experimental replicates for the earliest known positive cell type, for FACS and MS, respectively. Positive cell types were HSC for cKit and CD150, MPPa for CD34, MPPc for Flt3 and GMP for CD16/32. The percentage with respect to the average was calculated and graphed for each replicate. For MS replicates, t-tests were not conducted given variability in detection frequency across replicates.

## Total RNA content analysis

Four independent replicates were sorted, isolated, and quantified for analysis. Cells were purified as described above and 1000 cells of each cell type were sorted from four mice (two males, two females) into TRIzol. Cells lysed in TRIzol were heated to 37°C for 5 min to ensure complete homogenization. Phase separation was achieved by adding chloroform (100 µL), vortexing for 5 s, and then centrifugation at 12,000 g for 15 min at 4°C. Aqueous phases from each sample were carefully removed without disturbing the interphase. Three volumes of 100% ethanol were added to each aqueous phase to precipitate the RNA and vortexed for 5 s to mix. Precipitated RNA samples were added to Zymo-5 clean and concentrator columns and RNA was cleaned up as per the manufactures recommended protocol (Zymo Research, Tustin, California). RNA was finally eluted twice in 20 µL (final 40 µL) and quantification performed using a NanoDrop (ThermoFisher Scientific, Waltham, Massachusetts). Total RNA content was normalized with respect to relative cell size as determined by FSC-A analysis.

## Analyses with proteome only

### PANTHER 15.0 gene list analysis

Gene names of proteins detected by mass spectrometry for each cell type in at least one replicate were uploaded to the PANTHER gene list analysis website (http://www.pantherdb.org/) (*Mi et al., 2019a*; *Mi et al., 2019b*). The following parameters were selected for the search: List Type – ID List; Organism –*Mus musculus* Analysis – Functional classification viewed in graphic charts (bar chart). Ontology – Protein class. The data was exported for each cell type and the % of gene hits across total number of protein class hits graphed in GraphPad Prism (GraphPad, San Diego, California).

## PCA

Normalized data was further log-2 normalized by gene after adding 1/1000 of the global non-zero minimum value to account for zero values. PCA was performed using the pca package available from scikit-learn in python. The list of genes and their contribution to each of the two components is available in *Supplementary file 1*, Table 3 (young adult only) and *Supplementary file 1*, Table 6 (young adult with old mouse HSCs).

## Single sample gene set enrichment analysis

Using the msigdbr package available in R, C5:BP (GO biological process) gene sets were used for enrichment analysis (*Liberzon et al., 2015*; *Liberzon et al., 2011*; *Subramanian et al., 2005*). For each gene set, $log_2$ normalized data of individual replicates including young and old adult data were run through single sample gene set enrichment analysis (ssGSEA) available in R through the Gene Set Variation Analysis (GSVA) package (*Hänzelmann et al., 2013*). The enrichment scores were analyzed via Kruskal-Wallis test to determine if any differences existed between cell types for all gene

sets. Significance was determined using Benjamini-Hochberg procedure with FDR = 0.05. The stringency was further increased by only considering gene sets with at least 30 genes and at least half of the genes detected within the proteomic data. Of the 7526 gene sets available, 1108 gene sets were significant while meeting our criteria. All gene sets presented as bar graphs in the manuscript were within the 1108 gene sets. Bar graphs were plotted by finding the average enrichment score across replicates for each cell type, and ranked according to the average enrichment scores. In bar graphs without old adult cell types, enrichment analyses were still carried out including the old adult data. No additional pair-wise tests were done as we were ultimately interested in global differences and rankings between cell types.

### Unique minimum/zero values for ribosomal proteins

In subset of genes starting with 'Rpl' (Ribosomal protein large) or 'Rps' (Ribosomal protein small), the cell with the lowest intensity value or uniquely zero value was determined (genes with multiple zeros was classified as 'other'). For each cell type, the number of such ribosomal genes was counted and the result was plotted on a pie chart.

### Comparison between young and old adult HSC proteomes

Within proteins detected in at least three replicates in old HSCs, proteins that were either not expressed in young HSCs or were in the top 2.5% of old/young intensity fold-change values were identified as proteins differentially-detected (higher) in old compared to young HSCs.

### Analyses with proteome and transcriptome

Characterizing expression in transcriptome vs proteome for each cell type

For each cell type, the genes were classified as mRNA only or both. Within the mRNA only category, if a gene was uniquely detected as mRNA only in one particular cell type and detected as protein in at least three replicates in other cell types, the gene was considered a uniquely untranslated gene ('unique mRNA only'). Because such unique mRNA genes were so high in HSCs, we considered if there was enrichment beyond a stochastic distribution. To that end, we established a null distribution. For each cell type independently, all genes were randomly assigned the category of 'mRNA only' or 'both' with proportions equal to the distribution in the actual cell type. For each cell type, the number of unique mRNA only genes was determined as determined based on the random assignment using the same method described above. This was repeated 1000 times, and the 95% confidence interval for what the expected number of unique mRNA genes would be for each cell type was determined. The actual unique mRNA gene count was above the confidence interval for HSCs and below for MPPs.

Correlating proteomics to transcriptomics within cell type

To consider the degree of monotonic relationship between mRNA and protein, Spearman correlation was evaluated between proteomics and transcriptomics for each cell type using the cor.test function available in base R package. Only genes that were detected both in the transcriptome and proteome for each cell type were used.

Correlating proteomics to transcriptomics between cell types

To consider the relationship between protein/mRNA fold changes for each gene between cell types, we subset the $\log_2$ of the combined data on genes that were detected in both the transcriptome and proteome across all four cell types. Within this subset, Pearson correlation was calculated between permutations of cell types using the Pearson function available in base R package. The Pearson correlation values of fold changes between cell types were plotted on a heat map. To find the genes that were most differentially translated in all cell types compared to HSCs (specifically, highly translated in MPPs, lowly translated in HSCs), we compared the protein/mRNA fold-change values of HSC against each MPP for all genes detected by both cell types as mRNA and protein. For each MPP, the top 2.5% genes less translated in HSCs were determined. The intersection of such top 2.5% genes for comparisons to all MPPs was found. This list is available in **Supplementary file 1**, Table 9.

### miRNA target distribution

We downloaded the list of putative miRNA targets from mirdb.org (miRDB v6.0, source miRBase 22) (*Chen and Wang, 2020*; *Liu and Wang, 2019*). RefSeq IDs were converted to entrez IDs using the biomaRt package. The list was filtered for miRNAs pertaining only to *Mus musculus*. Within this list of genes, in evaluating the distribution of mRNA only, mRNA only unique to a cell type and both mRNA and protein, the same procedure was used as in characterizing expression in transcriptome vs proteome for each cell type, including the use of a null distribution.

## Quantification and statistical analyses

Cell-to-cell analyses were performed on GraphPad Prism. All other analyses were performed via Python 2.7.15 or R version 4.0.2. In Python, Numpy 1.15.3 was used for vector operations, scikit-learn 0.20.3 for dimensionality reduction and matplotlib 2.2.3 for graphing the PCA. In R, the packages GSVA 1.36.0 was used for ssGSEA, msigdbr 7.1.1 for loading gene sets from msigdb, biomaRt 2.44.1 for converting Enembl IDs to entrez IDs, ggfortify_0.4.10 for plotting dimensionality reductions, ggplot2_3.3.2 for making graphs of data analyzed in R, and base packages for statistical tests (spearman and pearson). When calculating fold changes, 0.00001 was added to all values in instances where zeros were part of the data. Kruskal-Wallis test and unpaired t-tests were used where appropriate. For enrichment analyses, significance was determined using Benjamini-Hochberg procedure with FDR = 0.05. Otherwise, for *Figure 4E* and *Figure 4—figure supplement 3*, significant genes were enriched by taking the top 2.5% of each list within an analysis, which was more stringent than assuming normality and using two standard deviations as a cutoff.

## Data code and availability

The datasets generated during this study are available as *Supplementary file 1*, Tables 1-10. The Python and R Code used for data compilation, analyses and graph-making is available at https://github.com/jnoh4/PofHemat.

## Acknowledgements

We thank Linda Quinn, Aaron McCarty, and Teja Naik for technical assistance. We thank Catherine Carswell-Crumpton, Patty Lovelace and Stephen Weber for flow cytometry assistance. We thank Carolyn Bertozzi for RAF's contribution to this manuscript. This study was supported by the California Institute for Regenerative Medicine RT3-07683, the Ludwig Cancer Foundation and NIH/NCI Outstanding Investigator Award R35CA220434 (to ILW); the Program in Translational and Experimental Hematology T32 from the National Heart, Lung, and Blood Institute T32HL120824 and the American Cancer Society Fellowship PF-15-142-01-CDD (to BWZ); PHS grant CA09302, awarded by the National Cancer Institute, DHHS (to BMG); the Damon Runyon Cancer Research Foundation Post-Doctoral Fellowship (to RAF); and the Leukemia and Lymphoma Society Special Fellows grant (to ACW).

## Additional information

### Funding

| Funder | Grant reference number | Author |
| --- | --- | --- |
| California Institute of Regenerative Medicine | RT3-07683 | Irving L Weissman |
| Virginia and D.K. Ludwig Fund for Cancer Research | | Irving L Weissman |
| NIH | R35CA220434 | Irving Weissman |
| National Heart, Lung, and Blood Institute | T32HL120824 | Balyn W Zaro |
| American Cancer Society | PF-15-142-01-CDD | Balyn W Zaro |
| National Cancer Institute | CA09302 | Benson George |

| Damon Runyon Cancer Research Foundation | Ryan A Flynn |
|---|---|
| Leukemia and Lymphoma Society | Adam C Wilkinson |
| American Society of Hematology | Victoria L Mascetti |

The funders had no role in study design, data collection and interpretation, or the decision to submit the work for publication.

## Author contributions

Balyn W Zaro, Conceptualization, Data curation, Formal analysis, Supervision, Funding acquisition, Validation, Investigation, Visualization, Methodology, Writing - original draft, Writing - review and editing; Joseph J Noh, Data curation, Formal analysis, Validation, Visualization, Methodology, Writing - original draft, Writing - review and editing; Victoria L Mascetti, Validation, Visualization, Methodology; Janos Demeter, Data curation, Formal analysis; Benson George, Conceptualization, Methodology, Writing - original draft; Monika Zukowska, Validation; Gunsagar S Gulati, Data curation, Formal analysis, Supervision; Rahul Sinha, Data curation, Formal analysis, Investigation, Methodology; Ryan A Flynn, Formal analysis, Investigation, Visualization; Allison Banuelos, Allison Zhang, Investigation; Adam C Wilkinson, Formal analysis, Supervision, Methodology, Writing - review and editing; Peter Jackson, Methodology, Writing - original draft; Irving L Weissman, Conceptualization, Supervision, Funding acquisition, Writing - original draft, Writing - review and editing

## Author ORCIDs

Balyn W Zaro (iD) https://orcid.org/0000-0002-8938-9889
Joseph J Noh (iD) https://orcid.org/0000-0003-3846-9771

## Ethics

Animal experimentation: Mice for all experiments were immunocompetent and group housed in an AAALAC certified barrier facility. The light cycle in the facility is 12h on/12h off. All experiments were performed according to guidelines established by the Stanford University Administrative Panel on Laboratory Animal Care under protocol #10266.

## Decision letter and Author response

Decision letter https://doi.org/10.7554/eLife.62210.sa1
Author response https://doi.org/10.7554/eLife.62210.sa2

# Additional files

## Supplementary files

• Supplementary file 1. Data tables. (1) Mass spectrometry individual runs for all cell types. (2) Mass spectrometry runs combined by cell type. (3) Contributions to the first two components of Principal Component Analysis (PCA) for young adult mass spectrometry data. (4) Proteins uniquely detected in select subsets of cell types. (5) Comparison of mass spectrometry data to data published by Cabezas-Wallscheid et al. (6) Contributions to the first two components of PCA for young and old adult mass spectrometry data. (7) Proteins either detected in old HSCs but not in young adult HSCs or within the top 2.5% of old/young fold-change in HSCs. (8) RNA-sequencing individual and combined runs for HSCs and MPPs. (9) Proteins uniquely decoupled from mRNA levels in HSCs compared to MPPs. (10) Number of overlaps between each miRNA's predicted target list with the list of proteins uniquely absent by protein but present by mRNA in HSCs compared to MPPs.

• Transparent reporting form

## Data availability

All code is available on GitHub and all raw and processed mass spectrometry data is available on the PRIDE database. Details are included in manuscript. Complete processed data available in searchable excel spreadsheet tables.

The following dataset was generated:

| Author(s) | Year | Dataset title | Dataset URL | Database and Identifier |
|---|---|---|---|---|
| Zaro BW, Noh JJ, Mascetti VL, Demeter J, George BM, Zukowska M, Gulati GS, Sinha R, Banuelos AM, Zhang A, Jackson PK, Weissman I | 2019 | Proteomic analysis of adult and aged mouse hematopoietic stem cells and their progenitors reveals post transcriptional regulation in stem cells | https://doi.org/10.6019/PXD017442 | PXD017442, 10.6019/PXD017442 |

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
