## [Decision Letter]

**Acceptance summary:**

Proteomic analyses on hematopoietic stem cells have been challenging due to limited material. This paper elegantly utilized state-of-the-art mass spectrometry and successfully scaled down the technique to obtain good quality proteomic data from small numbers of young and old adult mouse hematopoietic stem cells and progenitor cells. This paper provides a great resource to the field with broader implications for understanding mechanisms for stem cell maintenance, niche interactions, and fate determination.

**Decision letter after peer review:**

Thank you for submitting your article "Proteomic analysis of hematopoietic stem cells and progenitors reveals post transcriptional regulation in stem cells" for consideration by *eLife*. Your article has been reviewed by three peer reviewers, one of whom is a member of our Board of Reviewing Editors, and the evaluation has been overseen by Utpal Banerjee as the Senior Editor. The following individual involved in review of your submission has agreed to reveal their identity: Sean J Morrison (Reviewer #2).

The reviewers have discussed the reviews with one another and the Reviewing Editor has drafted this decision to help you prepare a revised submission.

Summary:

The authors performed proteomic profilings of mouse adult and aged HSCs, multipotent progenitors and oligopotent progenitors using mass spectrometry. The adult HSC compartment had decreased protein diversity compared to other compartments. Validation of differentially translated proteins revealed that Dnmt3a protein levels are undetected in adult HSCs, although its mRNA is expressed at levels comparable to those in MPPs. In addition, they identified a subset of genes with apparent post-transcriptional repression in adult HSCs, which included many miRNA target genes including Dnmt3a.

1) The authors showed that many putative miRNA target genes uniquely undetected at the protein level in adult HSCs are detected in aged HSCs (Figure 5—figure supplement 4). Are these alterations with aging associated with the reductions in miRNA expression?

2) The authors show in Figure 1C the number of proteins detected in each cell type (protein diversity). The authors should clarify the fraction of these proteins that overlapped between the different cell populations to highlight commonly expressed proteins and those that are more restricted in their expression.

3) In Figure 3A the authors report the total numbers of proteins identified in young and old HSCs. The authors should provide the identities of the proteins that were most differentially abundant between old and young HSCs. This would increase the impact of the manuscript given that this comparison is the most novel part of the paper.

4) Please remove speculative aspects of glycolysis and glutathione/ROS etc until a functional basis has been established (see #2 below in the section on expected future work).

5) The manuscript is well written in general with clear data presentation. However, many comparisons in mass spec data are without clear statistics (for example, Figure 2B, D, Figure 3C). I understand that zero-value data were not presented, possibly due to detection limit. But some statistics with the whole dataset will provide needed robustness to the conclusion.

6) In Figure 4C, the authors found poor correlation between protein and mRNA expression in HSCs and MPPs. Although the correlation was the lowest in HSCs, it was considerably low in MPPs as well (Spearman correlation 0.3 vs. ~0.4). Is it technical limitation of detecting proteins that result in the low correlations across all cell types?

7) The quality and size of the Figure 2E can be better. It is hard to see the expression.

Revisions expected in follow-up work:

1) The authors proposed the implication of miR29a in the compromised translation of Dnmt3a in HSCs. Is miR29a specifically expressed in HSCs? Please check its expression in HSCs, MPPs, and LPPs by RT-qPCR. Also, please show the protein levels of other target genes of miR29a in HSCs.

2) The discussion of glycolysis and glutathione metabolic processes in the Results comes across as being a little superficial. Although many claims have been made related to glycolysis in HSCs the reality is that nobody has done the isotope tracing in vivo that would be required to test these claims. Glycolytic flux cannot be predicted based on protein or RNA expression levels. Moreover, whether HSCs are highly glycolytic or not, it's not clear what implications this would have for ROS levels or glutathione homeostasis. Most importantly, the authors do not state what direction "glutathione metabolic process" proteins are changing in between HSCs and other cell types, which proteins are changing, or whether they are changing in the same direction.

---

## [Author Response]

Revisions for this paper:1) The authors showed that many putative miRNA target genes uniquely undetected at the protein level in adult HSCs are detected in aged HSCs (Figure 5—figure supplement 4). Are these alterations with aging associated with the reductions in miRNA expression?

We are fascinated by this possibility and thank the reviewer for their insight. To explore this possibility, we summarized the list of putative miRNA targets not detected in adult HSCs but detected in aged HSCs in Figure 5E where a large number of undetected proteins are now detected. While reduction in miRNA expression certainly can be one mechanism for the increased proteomic diversity in aged HSCs, there are also many other possibilities that may contribute to our observation. As of now, we would prefer not to speculate too much more without additional experiments, which are beyond the scope of our current manuscript. However, these are certainly experiments of interest to us and would serve as appropriate updates to our manuscript post-publication as per *eLife* policies.

2) The authors show in Figure 1C the number of proteins detected in each cell type (protein diversity). The authors should clarify the fraction of these proteins that overlapped between the different cell populations to highlight commonly expressed proteins and those that are more restricted in their expression.

We apologize that this data was not made obvious to the reviewers. Figure 2F includes the data of interest – 3130 proteins were detected across all cell types analyzed, and 619 proteins were uniquely absent in the HSC compartment. The number of proteins unique to each cell type are also included (ex. 340 proteins in GMPs). We have included additional discussion of this figure for added clarification, though we feel that it is currently located in the appropriate spot in the manuscript after validation of the quality of the dataset under the subheading “Characterization of proteins uniquely absent/detected by cell type(s)”. The manuscript now reads: *“*Over 40% of proteins were detected across all cell types, 3130 proteins in total (Figure 2F). […] For example, 619 proteins were absent in the HSC compartment but were found in all other cell types.”

3) In Figure 3A the authors report the total numbers of proteins identified in young and old HSCs. The authors should provide the identities of the proteins that were most differentially abundant between old and young HSCs. This would increase the impact of the manuscript given that this comparison is the most novel part of the paper.

We thank the reviewers for requesting this analysis. After filtering for proteins that were detected in at least three replicates in old HSCs, we generated a list of proteins (now Table 7, with Tables 7, 8, 9 shifted to 8, 9, 10 respectively) that were either 1) not detected in adult HSCs but detected in old HSCs or 2) in the top 2.5% of old/young intensity foldchange values. The table is now referenced in the section titled “Characterization of Old Adult HSC and MPP proteomes” with the methods used to identify the proteins included in the “Analyses with Proteome Only” section in the Materials and methods.

4) Please remove speculative aspects of glycolysis and glutathione/ROS etc until a functional basis has been established (see #2 below in the section on expected future work).

Since these data were shown as a validation of the mass spectrometry data rather than a key finding for the manuscript, we have chosen to remove a significant portion of the text related to glycolysis and GO enrichment analysis of glutathione metabolic process. Validation studies for Esam and Pfkl are now consolidated to a single paragraph. We hope this addresses the reviewers’ concerns related to glycolysis and glutathione metabolic process discussions and improves the focus of the manuscript.

5) The manuscript is well written in general with clear data presentation. However, many comparisons in mass spec data are without clear statistics (for example, Figure 2B, D, Figure 3C). I understand that zero-value data were not presented, possibly due to detection limit. But some statistics with the whole dataset will provide needed robustness to the conclusion.

We understand the reviewers’ request for statistics. Excluding statistics was a conscious decision we made while writing our manuscript for reasons we hope to explain here. We would first like to note that each individual mass spectrometry sample (MS) was analyzed and subjected to a false discovery rate cutoff of 1% in addition to the search parameters and statistics outlined in our deposited MS data. Therefore, each replicate already represents a statistically-significant readout. When visualizing the data and comparing between samples, we chose not to include statistics in part due to complications introduced by 0 intensity values. 0s introduce a significant amount of variability to a cell type’s protein detection profile. For example, a protein detected 4 times over 6 replicates with an intensity value of 100 could be undetected twice. Should that protein be assigned a value of 100 or 66.7? A parametric test assumes that a dataset of interest follows the distribution model posited by the statistical test (ex. t-distribution for t-test), but this assumption is challenged by the presence of 0s. On the other hand, a non-parametric test loses information on the quantitative nature of our dataset while treating all zeros equally (not all zero values may be equal especially between cell types and even from protein-to-protein depending on intensity of the protein when detected). We chose not to perform statistical tests after removing 0-values as that can also skew interpretation from protein to protein and cell type to cell type since the number of 0s can vary especially when a protein is lowly detectable. Given the challenges introduced by accounting for or omitting 0-values, we opted to present the data while keeping in mind the purpose of our manuscript – a tool for initial discovery followed up with orthogonal methods of validation such as FACS and microscopy. The best demonstration of such use is given by our example of Dnmt3a in our manuscript which we provide statistics for in our orthogonal validation. Additionally, we consulted several biostatisticians and have been advised that there is not a widely-accepted statistical method that can be applied across multiple, low-input proteomic datasets such as our entire dataset. The technical and experimental validation required to develop and apply a novel statistical model for our data is beyond the scope of this manuscript. We hope the reviewer will be understanding of this limitation. There are methods that we could use to provide some an error bar or a p-value, but we feel that they would detract from the presentation of our data as a platform for discovery. However, if the Editors and reviewer feel that the addition of a specific statistical analyses beyond what is already performed for each replicate is absolutely required for publication in *eLife*, we are willing to reconsider.

6) In Figure 4C, the authors found poor correlation between protein and mRNA expression in HSCs and MPPs. Although the correlation was the lowest in HSCs, it was considerably low in MPPs as well (Spearman correlation 0.3 vs. ~0.4). Is it technical limitation of detecting proteins that result in the low correlations across all cell types?

The correlations were calculated based on genes that were expressed as both mRNA and protein for each cell type, so the proteins that were not detected did not contribute to the correlation values. Nonetheless, the poor correlation between mRNA and protein in yeast and mammalian cells is well-documented in literature, so we are not surprised or concerned by our analysis (Gygi et al., 1999; Koussounadis et al., 2015; Liu et al., 2016). These references are included in the original manuscript. Additionally, Amon et al., 2019, which reports MS analysis of human HSCs and MPPs as a mixed population also observes a reduced correlation between mRNA and protein in the HSC/MPP compartment (R^2^=0.32) compared to committed progenitor populations MEP and GMP (R^2^ = 0.50 and 0.41, respectively). Our correlation between mRNA and protein in the HSC compartment is found to be 0.300, which is consistent with this finding. However, we are also careful in the text to emphasize that our protein datasets represent what is currently detectable by mass spectrometry analysis, and it is possible that additional proteins are present that cannot be detected due to extremely low abundance, although they will not significantly alter the correlation between mRNA and protein as we currently present in our manuscript. The text now reads:

"The correlation was lowest in the HSC compartment (ρ = 0.300), with comparable levels between MPPs (Figure 4B, C and Figure 4—figure supplement 1). Importantly, these correlation values are similar to what has been previously reported for a mixed population of human HSCs and MPPs (Amon et al., 2019)."

7) The quality and size of the Figure 2E can be better. It is hard to see the expression.

We have enlarged the figure considerably in order to make the staining differences for Dnmt3a and Ki67 between fresh HSCs and Cultured HSCs more apparent.

Revisions expected in follow-up work:1) The authors proposed the implication of miR29a in the compromised translation of Dnmt3a in HSCs. Is miR29a specifically expressed in HSCs? Please check its expression in HSCs, MPPs, and LPPs by RT-qPCR. Also, please show the protein levels of other target genes of miR29a in HSCs.

We apologize for not explicitly stating this in the manuscript and for not including the appropriate citation. We and others have previously reported that miR-29a is more highly expressed in HSCs compared to progenitor cells. The text now reads: “Notably, miR-29a has been shown the be highly expressed in HSCs compared to progenitor cells and has been implicated in negatively regulating Dnmt3a levels, in turn, promoting self-renewal (Han et al., 2010, Hu et al., 2015).”